# LARGE EEG-U-TRANSFORMER FOR TIME-STEP LEVEL DETECTION WITHOUT PRE-TRAINING

## ABSTRACT

Electroencephalography (EEG) reflects the brain's functional state, making it a crucial tool for diverse detection applications, including event-centric analysis like seizure detection and status-centric analysis like pathological detection. While deep learning-based approaches have recently shown promise for automated detection, traditional models are often constrained by limited learnable parameters and only achieve modest performance. In contrast, large foundation models showed improved capabilities by scaling up the model size, but required extensive time-consuming pre-training. Moreover, both types of existing methods focus on window-level classification, which requires redundant post-processing pipelines for event-centric tasks. In this work, based on the multi-scale nature of EEG events, we propose a simple U-shaped model to efficiently learn representations by capturing both local and global features using convolution and self-attentive modules for sequence-to-sequence modeling. Compared to other window-level classification models, our method directly outputs predictions at the time-step level, eliminating redundant overlapping inferences. Beyond sequence-to-sequence modeling, the architecture naturally extends to window-level classification by incorporating an attention-pooling layer. Such a paradigm shift and model design demonstrated promising efficiency improvement, cross-subject generalization, and state-of-the-art performance in various time-step and window-level classification tasks in the experiment. More impressively, our model showed the capability to be scaled up to the same level as existing large foundation models that have been extensively pre-trained over diverse datasets and outperforms them by solely using the downstream fine-tuning dataset.

## 1 INTRODUCTION

Electroencephalography (EEG) is a method to record an electrogram of the spontaneous electrical activity of the brain. Such recorded biosignals dynamically reveal the brain's functional state, making it an essential tool for studying brain activity. Among the EEG signal processing analysis, certain tasks, such as pathological detection, are status-centric that require predicting the class of an input signal window, while other tasks, such as seizure detection, are event-centric that aim to identify transitions from background noise to meaningful events. Traditionally, neurologists implement analysis by manually checking large numbers of multi-channel EEG signals. However, visual analysis is time-consuming and prone to subjectivity. Therefore, the automation of the detection of the underlying brain dynamics in EEG signals is significant to obtain fast and objective EEG analysis.

In recent years, deep learning models have demonstrated impressive abilities to capture the intricate dependencies within time series data, making them a powerful tool for EEG signal analysis over traditional manual and statistical methods (Zhu & Wang, 2023; Seeuws et al., 2024; Thuwajit et al., 2021; M. Shama et al., 2023; Tang et al., 2021). More recently, large foundation models that take advantage of self-supervised learning techniques have shown promising results in EEG analysis (Wang et al., 2024; Jiang et al., 2024; Yang et al., 2023a; Kostas et al., 2021). However, most existing work implements the classification task at a sliding window level, which involves segmenting a signal recording into distinct windows and predicting a label for each sample. Converting discrete predictions into continuous masking for event-centric tasks involves extensive post-processing, which departs from existing algorithms in simultaneous detection. In addition, while foundation models successfully scaled up their size, which, in turn, achieved impressive performace, through pre-training,

such a process requires diverse datasets and tremendous time and computation resources. Moreover, most existing biomedical signal processing research trains and evaluates models using formulated training and testing datasets that have a fixed sequence length. Such experimental settings and evaluation metrics do not fit with real-world requirements and often limit the model design, as different model architectures might benefit from different sequence lengths.

In contrast to window-level classification models, sequence-to-sequence modeling, a type of encoder-decoder architecture that maps an input sequence to an output sequence, provides a straightforward solution to avoid redundant post-processing steps through time-step-level classification. As the semantic information of time series data is mainly hidden in the temporal variance, U-Net (Ronneberger et al., 2015), a fully convolutional encoder-decoder network with skip connections that was originally designed for image segmentation, becomes a competitive backbone and has been widely used in the scientific field (Zhu & Beroza, 2019; Li & Guan, 2021; Chatzichristos et al., 2020; Seeuws et al., 2024; Perslev et al., 2019; Mukherjee et al., 2023; Pan et al., 2025; Wang & Li, 2024). However, the drawback of such models also stands out. Firstly, U-Net primarily operates within local receptive fields, making it difficult for U-Net to capture global features effectively. Beyond that, building up a U-Net requires stacking deeper layers, often leading to vanishing gradients and overfitting.

**Present work.** In this work, we proposed a training/inference framework for sequence-to-sequence EEG modeling to get rid of redundant overlapping inference. Such a framework significantly improved the real-world usage efficiency and achieved a 10-fold runtime improvement compared to window-level baselines. Subsequently, we propose a simple U-shaped architecture, comprising of convolutions and transformers with a self-attention mechanism, to be integrated into our framework. Such an architecture solved the mentioned drawbacks of the U-Net architecture and, compared to the pure-Transformer model, demonstrated memory usage efficiency and improved ability to exploit local structure with better temporal invariance. Beyond time-step level classification, we propose to use a simple attention-linear pooling layer to aggregate time-step embeddings for window-level classification, making it a unified solution for both event-centric and status-centric EEG analysis.

In the experiment, we evaluate the proposed model against both event-level and sample-level metrics in the event-centric task, namely, seizure detection, to reflect realistic clinical requirements (Dan et al., 2024; Beniczky et al., 2017); and benchmark our approach using standardized window-level datasets for sleep-stage classification and pathological detection to facilitate direct comparisons with baseline methods. Our model consistently outperforms existing algorithms across all tasks. In the event-centric task, compared to window-level baselines, our time-step classification model achieves a 10-fold runtime improvement. Further cross-dataset evaluation highlights the model's robustness and cross-subject generalization. More impressively, unlike several large foundation models in the baseline that require extensive pre-training across various EEG datasets, our method achieves state-of-the-art performance by solely using the downstream fine-tuning dataset without any pre-training process.

In summary, our contribution is listed as follows:

- We go beyond window-level representation and propose a training/inference framework to do sequence-to-sequence modeling. Such a framework can be easily adapted to the window-level classification, making it a unified solution for both event-centric and status-centric EEG analysis.
- We propose the EEG-U-Transformer to be integrated into such a unified framework and demonstrate the state-of-the-art performance in both types of tasks.
- We show that our model outperforms existing EEG foundation models that have been extensively pre-trained over diverse datasets by solely using the downstream fine-tuning dataset, revealing practical insights and an under-explored question on validating the pre-training's trade-off between the return and cost.

## 2 METHODOLOGY

### 2.1 PRELIMINARY

For continuous EEG waveforms, the training dataset is generated by segmenting the waveform into bags of uniform windows $\mathcal{D} = (\mathcal{X}, \mathcal{Y}) = \{(x_i, y_i) \mid i = 1, \ldots, N\}$. Each input window $x_i \in \mathbb{R}^{T \times K}$

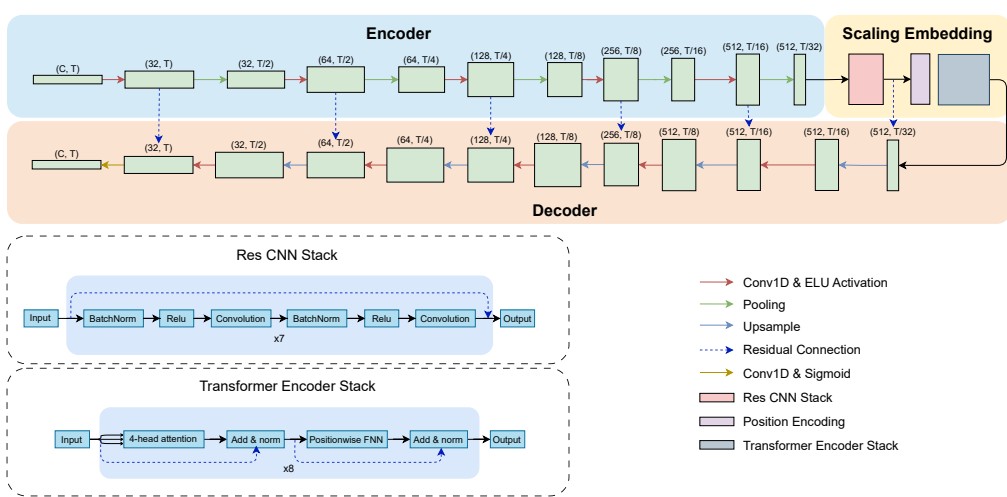

Figure 1: An example of our model's architecture in time-step level classification.

represents a multivariate time series with $K$ channels and $T$ time steps. We use $x_i[t, k]$ to denote the data value at time step $t$ and channel $k$ within the sample $x_i$. In window-level classification model, the ground truth label $y_i \in \{1, 2, \ldots, C\}$ indicates whether the window contains an activity, where C represents the number of classes. In contrast, in a time-step-level classification model, $y_i \in \{1, 2, \ldots, C\}^T$ is a box-shaped label set indicating the presence of an event at each time step. The model is trained to produce predictions $\hat{y}$ that minimize the classification objective, i.e., $\hat{y}_{:,i} = f_\theta(x_i)$, $\theta \in \arg\min \mathcal{L}$. Here, we use Cross-Entropy as our loss function $\mathcal{L}$, which, as shown in Equation 1, measures the dissimilarity between the predicted and true labels.

$$\mathcal{L}(y, \hat{y}) = -\frac{1}{T} \sum_{i}^{T} \sum_{j}^{C} y_{j,i} \, log(\hat{y}_{j,i}) \tag{1}$$

### 2.2 NETWORK DESIGN

At an intuitive level, we are motivated by the multi-scale nature of EEG events and design the neural network's architecture based on (1) convolution layers can efficiently down-sampling the long sequence and can exploit the local structure with a better temporal invariance, which, in turn, yields a better generalization; (2) self-attentive modules can help enriching the number of learnable parameters while integrating global information by the self-attention mechanism; and (3) a corresponding docoder is required to map high-level features back to the original length for time-step-level classification, thus to avoid redundant sliding window-level inference with high overlapping ratio. As a result, our model comes to be a U-shaped network with an encoder, a scaling embedding component, and a decoder, as shown in Fig. 1.

**Encoder.** The encoder comprises $N$ Convolution-MaxPooling blocks with various large kernel sizes, denoted as $K_s$, for each block, to comprehensively learn preliminary local features. Correspondingly, the padding parameter is set to be $\lfloor \frac{K_s}{2} \rfloor$ for each convolution layer. For each block, the input length will be down-sampled to half of its input size, and the feature dimension will be increased to the pre-defined out channel dimension. Essentially, after the encoder, the input signals were embedded into a preliminary vector representation $z \in \mathbb{R}^{d_{model} \times \frac{T}{2^N}}$, where the $d_{model}$ represents the final layer's output dimension.

**Scaling Embedding.** Inspired by Mousavi et al. (2020), after getting the encoded output, we implement a ResCNN stack (He et al., 2016) first to refine these tokenized features to yield a better generalization with better temporal invariance. The ResCNN stack consists of 7 blocks of Convolution-Convolution layers with residual connections. The output channel remains the same as the input, and the kernel size was set to be small($K_s \in \{2, 3\}$) to exploit local structure.

We then employ a transformer encoder stack Vaswani et al. (2017) to scale up the model size and to learn global representation across the tokenized signal. Specifically, the sine and cosine functions of different frequencies are used to be positional encodings,

$$PE_{(pos,2i)} = sin\left(\frac{pos}{10000^{\frac{2i}{d_{\text{model}}}}}\right), \quad PE_{(pos,2i+1)} = cos\left(\frac{pos}{10000^{\frac{2i}{d_{\text{model}}}}}\right)$$

which can then be summed with the input embedding. The refined representation, denoted as $Z$, will then be projected into equally-shaped query, key, and value spaces,

$$Q = ZW^Q, \quad K = ZW^K, \quad V = ZW^V,$$

and processed with the use of the global-attention mechanism as described in Equation 2.

$$A = softmax\left(\frac{QK^T}{\sqrt{d_k}}\right)V \tag{2}$$

The attention output is combined with tokens with a residual connection and layer normalization, and a subsequent feed-forward network to transform the output with another residual addition. Such hierarchical processing scales the model and integrates both local features and global context, enabling the model to learn complex temporal dependencies.

**Decoder.** Similar to the encoder, we use a convolutional decoder to decrypt the compressed information from the center latent space into a sequence of probability distributions. However, instead of the convolution-pooling block, we upsample the input with a scale factor of 2 and then with a convolution to decrease the number of channels and to increase the number of time steps back to the original window length. Residual connections are deployed between the encoder and decoder to facilitate efficient gradient flow.

Finally, the classifier was applied to project the time-step embedding into the targeted shape. For time-step level classification, the classifier is a simple one-dimensional convolution layer. For window-level classification, a learnable attention-pooling mechanism, described in Section 2.3, was applied to aggregate time-step representations.

## 2.3 ATTENTION POOLING

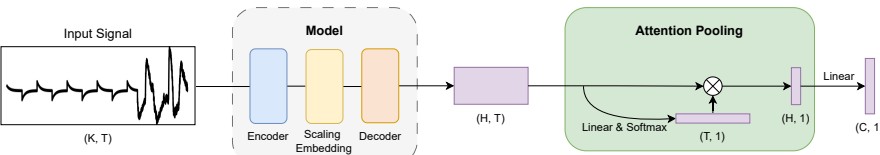

Figure 2: Attention-pooling layer for window-level classification.

To adapt the model to solve window-level classification tasks, we employ a learnable attention-based pooling mechanism, as shown in Figure 2, to efficiently embed each time step's high-level representations. Given the decoded feature map $Z \in \mathbb{R}^{H \times T}$, where $H$ denotes the output dimension of the final convolution layer in the decoder, we first compute a scalar attention score for each time step via a linear projection described in Equation 3, where $X_{perm} \in \mathbb{R}^{T \times H}$ is the transposed representation and $W_a \in \mathbb{R}^{H \times 1}$ is a learned parameter.

$$\mathbf{a} = \text{softmax}(\mathbf{W}_a \cdot \mathbf{X}_{perm}) \in \mathbb{R}^{T \times 1} \tag{3}$$

These attention weights are used to aggregate temporal features into a fixed-size context vector via weighted summation described in Equation 4.

$$\mathbf{z} = \sum_{t=1}^{T} a_t \mathbf{X}_{:,t} \in \mathbb{R}^{H \times 1} \tag{4}$$

This operation enables the model to selectively focus on informative temporal regions while remaining fully differentiable. The pooled representation $z$ is subsequently passed to a linear classifier followed by a sigmoid (or softmax) layer to produce the final window-level prediction.

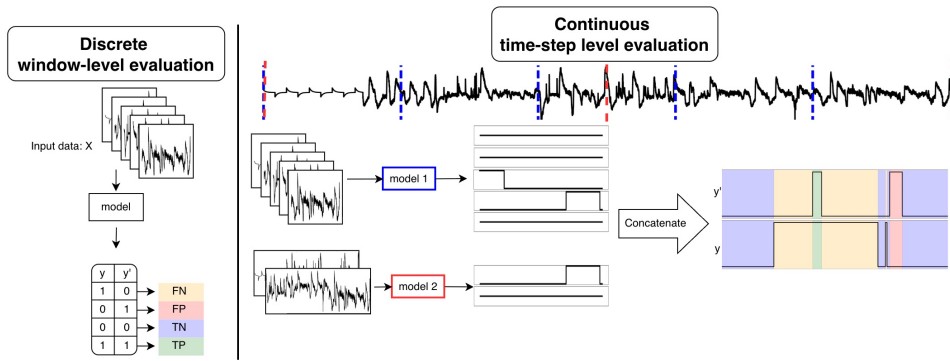

Figure 3: Two different inference and evaluation frameworks, where models can pick the best-matched sequence length under continuous evaluation.

## 2.4 SEQUENCE-TO-SEQUENCE MODELING

**Training.** Given a list of continuous EEG waveforms, a segmentation process was deployed to slice signals into uniform windows with fixed sequence length $T = D_{window} \times f_s$, where $D_{window}$ is the signal duration for a window in the unit of seconds, and $f_s$ denotes for the sampling frequency. We define a hyperparameter $r_{overlap}$, which represents the overlap ratio in time steps between consecutive windows, facilitating the augmentation of training samples.

The current EEG corpus poses a significant class imbalance challenge, as the majority of signals represent background activity, while meaningful events are sparsely distributed throughout the recordings. In our work, to enhance the model's capability to differentiate activity signals from background noise and other events, we statistically categorize training windows into three classes: no-activity, full-activity, and partial-activity, and uniformly sample a certain number of windows from each class to create a balanced dataset. Specifically, our training dataset is constructed as Equation 5.

$$\mathcal{D} = \mathcal{D}_{partial} \cup \mathcal{D}_{full}^* \cup \mathcal{D}_{bckg}^* \tag{5}$$

where $\mathcal{D}_{partial}$ comprises all partial-activity windows, while $\mathcal{D}_{full}^*$ and $\mathcal{D}_{bckg}^*$ are randomly selected subsets of full-activity and no-activity windows, respectively. The sizes of these subsets are determined by $|\mathcal{D}_{full}^*| = \alpha \times |\mathcal{D}_{partial}|$ and $|\mathcal{D}_{bckg}^*| = \beta \times |\mathcal{D}_{partial}|$. $\alpha$ and $\beta$ are weighting parameters controlling the relative proportions of windows full of events and windows lacking activities.

**Post-processing.** After having a sequence of probabilities outputted by the model, we implement a set of simple post-processing steps to convert continuous probabilities to the final detection. Initially, we apply a straightforward threshold filter to obtain a discrete mask as described in Equation 6, where the hyperparameter $\tau \in \mathbb{R}^c$ represents the threshold for each class.

$$\tilde{y}_i[t] = \begin{cases} c, & \text{if } \hat{y}_i[t, c] \geq \tau_c \\ 0, & \text{otherwise} \end{cases}, \quad \text{for } t = 1, \dots, T \tag{6}$$

Then, a pair of morphological operations, one with binary opening and one with binary closing operation, are employed using Virtanen et al. (2020) to eliminate spurious spikes of activity and to fill short 0 gaps. Lastly, we implement a simple duration-based rule to discard blocks of event labels lasting less than a minimal clinically relevant duration, denoted as $L_{min} = D_{min} \times f_s$, where $D_{min}$ represents the minimum duration seconds and $f_s$ represents the sampling frequency.

**Inference.** Traditionally, similar to the training set, the testing set in an experiment is a bag of fixed windows that are randomly sampled from the segmented patches. As described in the left part of Figure 3, window-level classification models directly perform inference over the formulated discrete input and evaluate over the corresponding ground truth labels.

In the context of this work, however, we elect to measure performance using a continuous time-step and event-level measure with the use of Dan et al. (2024). Specifically, time-step-based scoring

Table 1: Dataset/model statistics for each task.

| Task | Dataset | Class | Subject | Frequency | Model | Channel | Window length(s) |
|---|---|---|---|---|---|---|---|
| Sleep | Sleep-EDFx | 5 | 78 | 256 | All | 2 | 30 |
| Abnormal | TUAB | 2 | 253 | 200 | All | 23 | 10 |
| Seizure Detection | TUSZ | 2 | 675 | 256 | EEGWaveNet | 18 | 4 |
| | | | | | DCRNN | 19 | 57 |
| | | | | | Zhu-Transformer | 19 | 25 |
| | | | | | EventNet | 19 | 120 |
| | | | | | DeepSOZ-HEM | 19 | 600 |
| | | | | | Ours | 18 | 60 |

compares annotation labels time-step by time-step to detect TP, FP, TN, and FN. In contrast, event-based scoring assesses performance based on the temporal overlap between predicted and reference events. The detailed description of both scoring methods is available in the Appendix E. Such measures take a more holistic approach to evaluation and focus on the events in question, not on window-centric classification results.

We showed the continuous time-step level evaluation framework on the right side of Figure 3. Given a long continuous EEG waveform with activity masking, we firstly segment it into a sequential list of windows that match the model's input size. By popping windows from the queue and feeding them into the model, a sequence of masks will be output. Concatenating these sequential masks together will lead to the final annotation, which can then be compared with the ground truth label for the continuous recording under either time-step level or event level.

## 3 EXPERIMENT

### 3.1 SETTINGS

**Dataset.** We conduct experiments on one sequence-to-sequence modeling task, namely, seizure detection, and two window-level classification tasks, namely, sleep stage classification and pathological detection, to comprehensively evaluate the proposed method. For window-level tasks, we follow the experimental settings established in prior work, using standardized datasets with fixed channel counts and sequence lengths. In contrast, seizure detection is evaluated in a continuous manner, allowing the model to flexibly choose window shape. Dataset statistics for each task are summarized in Table 1, and descriptions are provided in Appendix H.

**Model implementation.** The ResCNN stack consists of seven residual blocks with kernel sizes $[3, 3, 3, 3, 2, 3, 2]$, each followed by batch normalization ($\epsilon = 10^{-3}$), ReLU activation, and spatial dropout. The transformer encoder contains 8 stacked layers with an embedding dimension of 512, 4 attention heads, and a feedforward dimension of 2048. The number of encoder and decoder blocks, as well as the filter and kernel size for their convolution layers, varies between different tasks. Detailed architecture is available in the Appendix G.

**Training.** We implemented our deep learning model using PyTorch and trained on 1 NVIDIA L40S 46GB GPU. For seizure detection, our training parameters include a batch size of 256, a learning rate of 1e-4, a weight decay of 2e-5, and a drop rate of 0.1 for all dropout layers. We use Binary Cross-Entropy loss as the objective function and RAdam as the optimizer. The training process was set to be 100 epochs with early stopping if no improvement in validation loss was observed over 12 epochs. For two window-level classification tasks, we use the same training configurations with EEGPT (Wang et al., 2024). We repeat the experiments five times with different random seeds.

### 3.2 TIME-STEP LEVEL CLASSIFICATION

Seizure detection is an event-oriented task, where epileptic seizures are the events of interest, which requires the model to output a set of $(t_{onset}, t_{duration})$ tuples in the SCORE compliant Beniczky et al. (2017), making it an ideal task for sequence-to-sequence modeling. We use Temple University Hospital EEG Seizure Corpus v2.0.3(TUSZ)Shah et al. (2018), the largest public dataset for seizure detection, to formulate our training dataset, and use its predefined testing recordings to evaluate model performance. The testing set is a list of blind EEG signals from different subjects that are

Table 2: Model performance in TUSZ's predefined testing set. The highest value is **bolded**.

| Evaluation Scale | Model | F1-score | Sensitivity | Precision |
|---|---|---|---|---|
| Sample-based | Gotman | 0.0679 | 0.0558 | 0.0868 |
| | EEGWaveNet | 0.1088 | 0.1051 | 0.1128 |
| | DCRNN | 0.1917 | 0.4777 | 0.1199 |
| | Zhu-Transformer | 0.4256 | 0.5406 | 0.3510 |
| | EventNet | 0.4830 | **0.5514** | 0.4286 |
| | DeepSOZ-HEM | 0.4466 | 0.4609 | 0.3791 |
| | Ours | **0.5730** | 0.4724 | **0.7281** |
| Event-based | Gotman | 0.2089 | 0.6199 | 0.1256 |
| | EEGWaveNet | 0.2603 | 0.4427 | 0.1844 |
| | DCRNN | 0.3262 | 0.5723 | 0.2281 |
| | Zhu-Transformer | 0.5387 | 0.6116 | 0.5259 |
| | EventNet | 0.5655 | 0.6116 | 0.5259 |
| | DeepSOZ-HEM | 0.5940 | 0.6222 | 0.4306 |
| | Ours | **0.6713** | **0.7168** | **0.6312** |

Table 3: Model's Runtime Over TUSZ's testing Set. The lowest runtime is **bolded**.

| Model | Total Runtime(s) | Runtime(s) per 1-hour EEG |
|---|---|---|
| DCRNN | 2571.75 | 60.24 |
| EEGWaveNet | 1690.19 | 39.59 |
| Zhu-Transformer | 3309.51 | 77.53 |
| Ours | **169.96** | **3.98** |

completely separated from the training set and validation set, which ensures the generalization of model performance.

We standardized the datasets used for training and testing by arranging 18 EEG channels in a consistent sequence, detailed discussed in Appendix H.1. The sequence length of a window is set to be 1-minute, sampled at 256 Hz, i.e., $T = 15360$. In dataset formulation, followed by Equation 5, we use $\alpha = 0.54$ and $\beta = 1.0$ to sample windows. The $r_{overlap} = 0.75$ is set to augment training samples, and $r_{overlap} = 0$ is used during the inference time. In post-processing, we adjust hyper-parameters based on the validation set's performance. Specifically, threshold $\tau$ was set to $0.8$ and minimum seizure duration $D_{min} = 2$. Detailed hyper-paremeter analysis are provided in the Appendix D.

**Baselines.** We implement one rule-based algorithm, namely, Gotman (Gotman, 1982), and five deep learning models, namely, Eventnet (Seeuws et al., 2024), Zhu-Transformer (Zhu & Wang, 2023), DCRNN (Tang et al., 2021), DeepSOZ-HEM(M. Shama et al., 2023), and EEGWaveNet (Thuwajit et al., 2021) with the use of SZCORE Dan et al. (2024). Every baseline's training dataset is formulated using the TUSZ's predefined training set, but different sampling strategies, input window lengths, and pre-processing processes are used.

**Evaluation Metrics.** We evaluate our method and baselines' F1-score, sensitivity, and precision with the use of the SZCORE framework (Dan et al., 2024) under the sample(time-step) and event scale as described in Section 2.4. The detailed description of both scale are provided in Appendix E.

As shown in Table 2, our model significantly outperforms other models under both evaluations by the improvement of $13.01\%$ under time-step level and $18.63\%$ under event level in terms of F1-score. It is noteworthy that we tune the post-processing threshold on the event-based performance, which leads to a relatively low sample-based sensitivity, but with a high precision. We also evaluate the sample-level AUROC distribution across testing waveforms to score models' performance without the impact of the threshold hyperparameter in the Appendix C.2.

**Runtime Analysis.** We further verify our model's efficiency by comparing the inference time, from the time that data was passed into the model to the time that the annotation file with HED-SCORE compliant was output, with other window-level classification models using TUSZ's testing set in Table 3. Our model demonstrates the lowest running time with the ability to handle a one-hour-long recording in 3.98 seconds. Compared to EEGWaveNet, our model achieves about 10-fold runtime improvement.

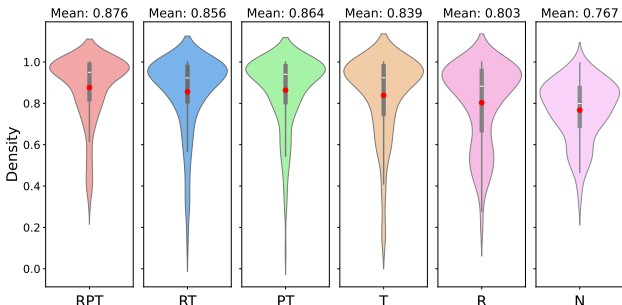

Figure 4: Ablation study for our model by evaluating the AUROC distributions. **N** represents a vanilla deep U-Net without ResCNN and Transformer encoder stack; **R** represents the U-Net with ResCNN stack; **T** represents the U-Net with Transformer Stack; **P** means adding positional encoding before feeding into the transformer stack.

Table 4: The classification performance on the Sleep-EDFx dataset. The highest value is **bolded**.

| Methods | Model Size | Balanced Accuracy | Cohen's Kappa | Weighted F1 |
|---|---|---|---|---|
| U-Sleep (Perslev et al., 2021) | 3.1M | $0.6720 \pm 0.0043$ | $0.6157 \pm 0.013$ | $0.7150 \pm 0.012$ |
| BENDR (Kostas et al., 2021) | 3.9M | $0.6655 \pm 0.0043$ | $0.6659 \pm 0.0043$ | $0.7507 \pm 0.0029$ |
| BIOT (Yang et al., 2023a) | 3.2M | $0.6622 \pm 0.0013$ | $0.6461 \pm 0.0017$ | $0.7415 \pm 0.0010$ |
| LaBraM (Jiang et al., 2024) | 5.8M | $0.6771 \pm 0.0022$ | $0.6710 \pm 0.0006$ | $0.7592 \pm 0.0005$ |
| EEGPT (Wang et al., 2024) | 25M | $0.6917 \pm 0.0069$ | $0.6857 \pm 0.0019$ | $0.7654 \pm 0.0023$ |
| Ours | 6.1M | $\mathbf{0.7201 \pm 0.0059}$ | $\mathbf{0.6954 \pm 0.0004}$ | $\mathbf{0.7693 \pm 0.0029}$ |

**Ablation Study.** We show each model component's necessity by testing multiple partial models after removing certain components. We use AUROC-distribution across testing recording files to ignore the impact of post-processing. As shown in Figure 4, vanilla U-Net has an underwhelming performance with a low AUROC mean. Solely adding a ResCNN stack or a transformer stack will marginally improve the model performance, but also lead to a bigger variance with some extreme false cases. By contrast, integrating both the ResCNN and Transformer stacks produces not only higher mean AUROC but also reduced variance, indicating that these components complement each other effectively. These results underscore the importance of each proposed element in achieving robust and accurate seizure detection.

### 3.3 WINDOW LEVEL CLASSIFICATION

We further validate the effectiveness of our model by following the most recent work's setting to conduct comparative experiments with state-of-the-art large EEG foundation models over window-level tasks, including stage classification for multi-class classification task and pathological detection for binary classification task.

**Sleep stage classification.** Following the EEGPT (Wang et al., 2024), we use Sleep-EDFx (Kemp et al., 2000) to formulate the training and testing datasets contain bags of windows with 2 channels and 30-second length sampled at 256Hz, i.e., $T = 7680$. *Balanced Accuracy, Weighted F1, and Cohen's Kappa* are used as evaluation metrics. For large foundation model baselines, we use the pre-trained weights as initialization and either fully fine-tune the model or train additional layers using the linear-probing method, based on the proposed work, over the downstream training set. In comparison, our model is directly trained over the downstream training set.

As shown in Table 4, our model exhibited accuracy improvements of $4.11\%$. Remarkably, with the use of convolution networks that can efficiently encode from temporal information, our model is able to scale up to the same level of size as other large foundation models while keeping a smooth gradient flow and effectively leading to a convergence without any pre-training. At the same time, such a convolution-transformer combination also outperforms the large pre-trained models with considerably more learnable parameters than our method, like EEGPT.

Table 5: The results of different methods on TUAB. The highest value is **bolded**.

| Methods | Model Size | Balanced Accuracy | AUROC |
|---|---|---|---|
| SPaRCNet (Jing et al., 2023) | 0.79M | $0.7896 \pm 0.0018$ | $0.8676 \pm 0.0012$ |
| ContraWR (Yang et al., 2023b) | 1.6M | $0.7746 \pm 0.0041$ | $0.8456 \pm 0.0074$ |
| FFCL (Li et al., 2022) | 2.4M | $0.7848 \pm 0.0038$ | $0.8569 \pm 0.0051$ |
| CNN-T (Peh et al., 2022) | 3.2M | $0.7777 \pm 0.0022$ | $0.8461 \pm 0.0013$ |
| BIOT (Yang et al., 2023a) | 3.2M | $0.7959 \pm 0.0057$ | $\mathbf{0.8815 \pm 0.0043}$ |
| ST-T (Song et al., 2021) | 3.5M | $0.7966 \pm 0.0023$ | $0.8707 \pm 0.0019$ |
| EEGPT (Wang et al., 2024) | 25M | $0.7983 \pm 0.0030$ | $0.8718 \pm 0.0050$ |
| Ours | 7.3M | $\mathbf{0.8144 \pm 0.0002}$ | $0.8568 \pm 0.0019$ |

**Pathological Detection.** We use TUAB (Shah et al., 2018), a corpus of EEGs that have been annotated as clinically normal(non-pathological) or abnormal(pathological). For the data splitting and baselines implementation, we strictly follow the same configuration as BIOT (Yang et al., 2023a) to fairly compare all methods. A window in the dataset contains 23 channels with 10-second signals sampled at 200Hz, i.e., $T = 2000$. Every baseline is a fully fine-tuned model, while, similar to sleep stage classification, we purely use the downstream dataset to train the model. Followed by EEGPT (Wang et al., 2024), the *Balanced Accuracy and AUROC* are used as evaluation metrics. We should acknowledge that we removed LabraM from the baseline list as this model is pre-trained on the TUEG dataset, which is a superset corpus of TUAB with a similar signal distribution.

The results are provided in Table 5, where our model achieves the best balanced accuracy with an improvement of $2.02\%$ over EEGPT. On the other hand, our method achieved the top-tier AUROC performance but was lower than the best model, BIOT. This might be led by the small sequence length, which, after the convolution layers, will output short feature vectors that, in turn, limit the global attention's performance.

### 3.4 CROSS-DEVICE SEIZURE DETECTION

Beyond testing under the same data distribution, our method demonstrated good cross-subject and cross-device generalization performance, which underscores its potential for real-world applications across patients and hospitals. In Table 6, we trained a model with the use of TUSZ's predefined training set and Siena Scalp dataset (Detti, 2020) and apply the algorithm to two different EEG corpus, namely, SeizeIT1(Vandecasteele et al., 2020) and Dianalund(Dan et al., 2024). Both datasets are acquired at different regions, from different subjects with varying ranges of age, and from different monitoring devices, leading to unique signal attributes that depart from the training set. The detailed dataset description is available in the Appendix.

Compared to CA-EEGWaveNet[1] and DeepSOZ-HEM, our model achieves the best event-based F1-score across both datasets($0.4547$ on SeizeIT1 and $0.4283$ on Dianalund). Although the evaluation results inevitably dropped, our model demonstrates the most stable performance(with $34\%$ F1-score drop, which is lower than DeepSOZ-HEM with a $47\%$ drop) across different data distributions, underscoring its generalization ability. The full cross-dataset performance leaderboard are provided in C.1

### 4 CONCLUSION AND DISCUSSIONS

In this paper, we propose to learn the EEG signal's representation at a time-step level to boost the EEG model's efficiency on event-centric tasks, such as seizure detection, by getting rid of redundant over-lapping inference and complicated post-processing steps. Beyond sequence-to-sequence modeling, our experimental results revealed that strong performance can be achieved through a well-designed, simple architecture without reliance on complex pre-training or massive data resources. Such results significantly lowered the barrier to deployment in clinical real-world settings.

While our model already achieves state-of-the-art performance on three downstream tasks, the proposed encoder-decoder architecture also supports a variety of pre-training strategies, such as

---

[1]A variation of the EEGWaveNet model proposed by IBM. Source code is available on `https://github.com/IBM/channel-adaptive-eeg-classifier`.

Table 6: Model performance in SeizeIT1 and Dianalund datasets. The highest value is **bolded**.

| Dataset | Evaluation Scale | Model | F1-score | Sensitivity | Precision |
|---------|------------------|-------|----------|-------------|-----------|
| SeizeIT1 | Sample-based | CA-EEGWaveNet | 0.0043 | 0.0007 | 0.0072 |
| | | DeepSOZ-HEM | 0.2211 | 0.3853 | 0.2274 |
| | | Ours | **0.2821** | **0.1615** | **0.6372** |
| | Event-based | CA-EEGWaveNet | 0.0200 | 0.0030 | 0.0500 |
| | | DeepSOZ-HEM | 0.2455 | 0.4686 | 0.2376 |
| | | Ours | **0.4547** | **0.3751** | **0.5623** |
| Dianalund | Sample-based | CA-EEGWaveNet | 0.0274 | 0.0070 | 0.2129 |
| | | DeepSOZ-HEM | **0.2870** | **0.4519** | 0.2805 |
| | | Ours | 0.2282 | 0.1240 | **0.4895** |
| | Event-based | CA-EEGWaveNet | 0.1437 | 0.0571 | 0.2000 |
| | | DeepSOZ-HEM | 0.3125 | **0.5844** | 0.2657 |
| | | Ours | **0.4283** | 0.3692 | **0.4488** |

Masked Autoencoders(MAE) (He et al., 2022). It is worth exploring the architecture's capability of unsupervised representation learning to further improve the classification performance in downstream tasks.

## ETHICS STATEMENT

In this study, we used multiple publicly available datasets that comply with medical ethical policies (e.g., the Declaration of Helsinki) and were approved by the Institutional Review Boards (IRBs) of the respective institutions that collected and shared the data. Therefore, ethical and privacy standards are assumed to be upheld.

Demographic information across datasets is often limited, and critical metadata such as medication usage or comorbidities is generally missing. In the absence of such information required for patient stratification, this and related work are limited to a one-size-fits-all modeling approach. This limitation could introduce bias during training and risk misdiagnosis during testing. Seizure detection can be performed offline (retrospective analysis) or online (real-time monitoring). In both cases, false positives (FPs) and false negatives (FNs) are significant, but in real-time settings, FNs are especially critical as they may delay medical intervention. Therefore, we strongly advocate for a human-in-the-loop framework when deploying such models in clinical environments, even if models are fine-tuned for patient-specific stratification.

## REPRODUCIBILITY STATEMENT

Our source code and model are available at `https://anonymous.4open.science/r/EEG-U-Transformer-5E86`, where a detailed README file has been provided for reproducing experiments. A detailed dataset description is also provided in Appendix H.

## LLM USAGE

We utilize LLMs (e.g., ChatGPT) to assist with language polishing during writing.

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

## A    RELATED WORK

**Automated EEG Analysis.** To mitigate the subjectivity and intensive manual effort associated with analyzing EEG data, various deep learning approaches have been employed, including models based on convolutional neural networks (CNNs), recurrent neural networks (RNNs), graph neural networks (GNNs), and Transformers.

CNN-based models Thuwajit et al. (2021); Wu et al. (2022) excel at extracting local spatial features but typically struggle with long-term temporal dependencies. Conversely, RNN-based models Abdelhameed et al. (2018); Saqib et al. (2020), especially those using Long Short-Term Memory (LSTM) architectures, effectively model temporal sequences but often encounter difficulties in spatial feature extraction and suffer from gradient vanishing issues over long sequences. Prior studies have combined CNN and RNN modules to simultaneously capture both spatial and temporal EEG features. Graph neural network (GNN) models Tang et al. (2021) approach EEG data as spatio-temporal graphs, extracting relational information among channels for tasks such as anomaly detection and classification. Transformer-based models, benefiting from recent advances in large language models, have emerged as robust tools for modeling long-term temporal dependencies. Similar to earlier efforts with RNNs, recent studies Li et al. (2020) combined CNN modules with Transformer components, aiming to leverage both spatial and temporal features inherent in multi-channel EEG data. Additionally, several works explored Global-local interaction and fusion Zhou et al. (2024); Zhao et al. (2024); Lou et al. (2025); Pan et al. (2022), which is similar to our model architecture, but only focused on either event-centric or status-centric tasks instead of proposing a unified framework.

Although these models have demonstrated impressive classification performance, the transition from window-level predictions to sample-level masks, indicating precise event onset times and durations, remains redundant and time-consuming. Additionally, most existing studies rely on window-level evaluation metrics, comparing predictions directly with ground truth labels per window rather than employing more clinically relevant event-level measures.

**Large Foundation Models.** With the success of Large Language Model(LLM) Brown et al. (2020); Devlin et al. (2019), more and more EEG research is focusing on building large foundation models. Such foundation models take advantage of a self-supervised learning strategy to learn the representation of EEG signals from a wide range of datasets. The pre-trained model is then adapted, either by fully fine-tuning or by probing from the outputted representation, to do various downstream tasks. For instance, BIOT Yang et al. (2023a) use a contrastive learning strategy to learn embeddings for biosignals with various formats; LaBraM Jiang et al. (2024) learns universal embeddings through a masked autoencoder to do unsupervised pre-training over 2500 hours of EEG data; EEGPT Wang et al. (2024) employs a dual self-supervised approach for pretraining, involving spatio-temporal representation alignment and mask-based reconstruction. Such models demonstrate impressive performance over a variety of downstream tasks while requiring extensive time and memory to do pre-training.

**U-Net.** U-Net Ronneberger et al. (2015) architecture was first proposed in the field of CV for image segmentation tasks. Considering the temporal continuity of time series data, such networks have been widely deployed in various sequence-to-sequence scientific signal processing applications, such as seismic phase detection Zhu & Beroza (2019), sleep-staging classification Li & Guan (2021); Perslev et al. (2019), denoising heart sound signals Mukherjee et al. (2023), and seizure detection Islam et al. (2023); Seeuws et al. (2024).

There are some works exploring combining U-Net with Transformer for other fields. For example, in a medical image segmentation task, Petit et al. (2021) used self and cross-attention with U-Net; Lin et al. (2022) incorporated hierarchical Swin Transformer into U-Net to extract both coarse and fine-grained feature representations. In seismic analysis, Mousavi et al. (2020) proposed a deep neural network that can be regarded as a U-Net with global and self-attention but without a residual connection. However, in the biomedical signal processing area, to the best of our knowledge, there is no existing work to scale U-Net using transformer blocks. The closest work to this paper is Islam et al. (2023), where multiple attention-gated U-Nets are used and a following LSTM network is implemented to fuse results.

Table 7: Model Performance in cross-device seizure detection task with the use of Dianalund dataset.

| Model | Architecture | Result | | |
|---|---|---|---|---|
| | | F1-score | Sensitivity | Precision |
| SeizureTransformer | U-Net & CNN & Transformer | 0.43 | 0.37 | 0.45 |
| Van Gogh Detector | CNN & Transformer | 0.36 | 0.39 | 0.42 |
| S4Seizure | S4 | 0.34 | 0.30 | 0.42 |
| DeepSOZ-HEM | LSTM & Transformer | 0.31 | 0.58 | 0.27 |
| HySEIZa | Hyena-Hierarchy & CNN | 0.26 | 0.6 | 0.22 |
| Zhu-Transformer | CNN & Transformer | 0.20 | 0.46 | 0.16 |
| SeizUnet | U-Net & LSTM | 0.19 | 0.16 | 0.20 |
| Channel-adaptive | CNN | 0.14 | 0.06 | 0.20 |
| EventNet | U-Net | 0.14 | 0.6 | 0.09 |
| Gradient Boost | Gradient Boosted Trees | 0.07 | 0.15 | 0.09 |
| DynSD | LSTM | 0.06 | 0.55 | 0.04 |
| Random Forest | Random Forest | 0.06 | 0.05 | 0.07 |
| SD2025 | LaBraM | 0.477 | 0.6975 | 0.0271 |
| NE Illusion | JEPA | 0.0269 | 0.6888 | 0.0157 |
| STORM | BENDR | 0.0203 | 0.9887 | 0.0112 |

## B CODE AVAILABILITY

Our source code and model are available at `https://anonymous.4open.science/r/EEG-U-Transformer-5E86`.

## C MORE RESULTS

### C.1 DIANALUND CROSS-DEVICE SEIZURE DETECTION

We provided the full model performance leaderboard in Table 7, where models are evaluated over the Dianalund dataset for cross-device seizure detection. As shown in the Table, our model achieves the best performance wih the highest F1-score.

### C.2 AUROC FOR SEIZURE DETECTION

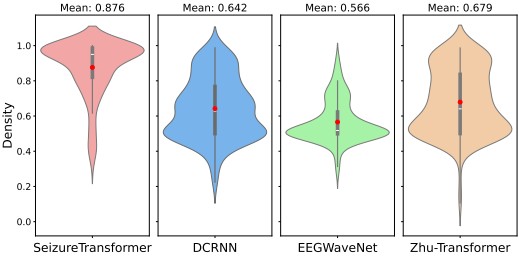

Figure 5: Violin plots illustrating the distribution of AUROC values for SeizureTransformer, DCRNN, EEGWaveNet, and Zhu-Transformer models evaluated on the TUSZ v2.0.3 predefined testing set. Mean AUROC scores for each model are indicated above each plot, with the SeizureTransformer demonstrating the highest overall performance.

We quantify the model's performance using the area under the receiver operating characteristic (AUROC). For each continuous EEG recording, the ROC curve plots the true and false positive rates across all possible decision thresholds, and the AUC represents the area under the ROC curve, which summarizes the model's performance. As shown in Figure 5, our model demonstrated the highest performance, with a mean AUROC of 0.876 and a distribution tightly concentrated toward higher values.

Table 8: Effect of dataset formulation (event-level $F_1$) with $\tau = 0.8$

| $\beta \backslash \alpha$ | 0.2 | 0.4 | 0.54 (full) |
|---|---|---|---|
| 1.0 | 0.6240 | 0.6554 | **0.6713** |
| 2.0 | 0.6638 | 0.6531 | 0.6667 |

Table 9: Threshold–performance trade-off (event-level metrics)

| $\tau$ | $F_1$ | Sensitivity | Precision |
|---|---|---|---|
| 0.9 | **0.6916** | 0.6549 | **0.7327** |
| 0.8 | 0.6713 | 0.7168 | 0.6312 |
| 0.6 | 0.6308 | 0.7611 | 0.5386 |
| 0.4 | 0.5914 | 0.7876 | 0.4734 |
| 0.2 | 0.5470 | **0.8053** | 0.4143 |

## D  HYPER-PARAMETER ANALYSIS

**Dataset formulation.**    Table 8 indicates that a balanced dataset, which is formulated with a balanced number of windows that lack seizures, full of seizures, and a partial part of seizures, will benefit the model's performance. Too many windows that are full of seizure activities will negatively influence the training quality. In that case, we recommend a balanced dataset, i.e., $\alpha = 1$ and $\beta = 1$, if conditions allow, to maximize the model's performance.

**Detection threshold.**    Table 9 reports the effect of varying the decision threshold $\tau$. Higher thresholds favor precision at the expense of sensitivity, whereas lower thresholds improve sensitivity but increase false positives. In practice, $\tau$ can be tuned to clinical priorities: intensive care monitoring may prioritize sensitivity, while wearable devices benefit from higher precision to reduce alarm fatigue.

Table 10: Model size (number of learnable parameters) vs. feed-forward width ($\dim_{\text{ff}}$) and number of encoder layers ($\text{num\_layers}$).

| $\dim_{\text{ff}} \backslash \text{num\_layers}$ | 4 | 8 | 12 |
|---|---|---|---|
| 1024 | 23,143,393 | 31,554,529 | 39,965,665 |
| 2048 | 28,391,393 | 41,000,929 | 53,610,465 |
| 4096 | 38,887,393 | 59,893,729 | 80,900,065 |

Table 12: Ablation of post-processing (morphological filtering and short-event removal).

| Post-process | $F_1$ | Sensitivity | Precision |
|---|---|---|---|
| normal | **0.6916** | 0.6549 | 0.7327 |
| no morphological filter | 0.6887 | 0.6460 | **0.7374** |
| no remove event | 0.6686 | 0.6785 | 0.6590 |
| none | 0.6156 | **0.7345** | 0.5298 |

**Model size scaling.**    As shown in Table 10 and 11, our proposed model fits the scaling law that model performance is better when the model size is scaled up(whether through the number of layers or the feedforward dimension). On the other hand, given the limited dataset size, the performance will drop when the model size is too big.

**Post-processing.**    We report the ablation study of post-processing in Table 12. Removing both steps slightly increases sensitivity but substantially hurts precision and overall $F_1$, showing that morphological filtering and short-event removal effectively reduce false alarms.

Table 11: Event-level $F_1$ vs. feed-forward width ($\dim_{\text{ff}}$) and number of encoder layers (num_layers).

| $\dim_{\text{ff}} \backslash$ num_layers | 4 | 8 | 12 |
|---|---|---|---|
| 1024 | 0.6626 | 0.6398 | 0.6838 |
| 2048 | 0.6598 | **0.6916** | 0.6760 |
| 4096 | 0.6740 | 0.6902 | 0.6568 |

# E  SAMPLE AND EVENT-BASED SCORING

These two scale scoring methods are proposed by Dan et al. (2024), which aims to align EEG machine learning research with real-world clinical requirements.

**Sample(Time-Step)-based Scoring.**  Annotations are evaluated at 1 Hz, aligning with human annotator resolution. Each 1-second sample is labeled as a true positive (TP), false positive (FP), or false negative (FN). Machine learning models can use arbitrary window sizes and overlaps, as long as they produce predictions at 1 Hz. For partial overlaps with seizures, a sample is labeled as "seizure" if the overlap exceeds $50\%$.

**Event-based Scoring.**  evaluates activities based on the overlap between predicted and reference events. Any overlapping prediction is counted as a true positive (TP), while non-overlapping predictions are false positives (FP).

Due to challenges in precisely annotating events' start and end times, caused by gradual EEG transitions and artifacts, some tolerance is introduced as follows for seizure detection evaluation:

- **Pre-ictal tolerance**: Predictions up to 30 seconds before the annotated onset are accepted.

- **Post-ictal tolerance**: Predictions up to 60 seconds after the annotated offset are accepted.

- **Minimum overlap**: Any overlap, however brief, is sufficient for detection.

- **Event merging**: Events separated by less than 90 seconds are merged, which corresponds to the combined pre- and post-ictal tolerance.

- **Maximum event duration**: Events longer than 5 minutes are split.

# F  PSEUDOCODE FOR TIME-STEP LEVEL CLASSIFICATION

---

**Algorithm 1** Window level classification model's prediction workflow at a time-step level classification.

---

**Require:** recording duration $T$ (in seconds), sampling frequency $f_s$, window size $D_{window}$ (in seconds), overlap ratio $r_{overlap} \in [0, 1)$, decision threshold $\tau$.
**Ensure:** Sample-level prediction mask $y_{\text{mask}}$.
1: $B \leftarrow \lfloor \frac{T - D_{window}}{(1 - r_{overlap}) \times D_{window}} \rfloor + 1$           {Number of windows to predict}
2: Predict $\hat{y} \in \mathbb{R}^B$
3: Initialize $y_{\text{mask}} \leftarrow \mathbf{0}^{T \cdot f_s}$
4: $w_s \leftarrow D_{window} \cdot f_s$           {Number of samples per window}
5: $s_s \leftarrow (1 - r_{\text{overlap}}) \cdot w_s$           {Window step size in samples}
6: **for** each index $i = D_{window}$ to $\text{len}(\hat{y}) - 1$ **do**
7:     $l \leftarrow \max(0, \lfloor i \cdot s_s \rfloor)$
8:     $r \leftarrow \min(T \cdot f_s, \lfloor (i + 1) \cdot s_s \rfloor)$
9:     $v \leftarrow \text{mean}(\hat{y}[\max(0, i - D_{window}) : \min(i, T)]) > \tau$
10:     $y_{\text{mask}}[l : r] \leftarrow \text{int}(v)$
11: **end for**
12: **return** $y_{\text{mask}}$

---

---

**Algorithm 2** Our model's prediction workflow at a time-step level classification.

---

**Require:** recording duration $T$ (in seconds), sampling frequency $f_s$, window size $D_{window}$, decision threshold $\tau$.

**Ensure:** Binary prediction mask $y_{\text{mask}} \in \{0,1\}^{T \times f_s}$.

1: $B = \lfloor \frac{T}{D_{window}} \rfloor$ {Number of samples per window}

2: Predict $\hat{y} \in \mathbb{R}^{D_{window} \cdot f_s}$

3: $\hat{y} \leftarrow \text{Flatten}(y_{\hat{y}})[: T \times f_s]$

4: $y_{\text{mask}}[t] \leftarrow \begin{cases} 1, & \text{if } \hat{y}[t] > \tau \\ 0, & \text{otherwise} \end{cases}$

5: # Other post-processing with a time complexity of $\mathcal{O}(D_{window} \cdot f_s)$

6: **return** $y_{\text{mask}}$

---

**Computational complexity.** Let a recording have duration $T$ seconds sampled at $f_s$ Hz, window size $D_{\text{window}}$ (in seconds), and overlap ratio $r_{\text{overlap}} \in [0,1)$. The stride (in samples) is

$$s_s \;=\; (1 - r_{\text{overlap}})\, D_{\text{window}}\, f_s,$$

which induces

$$B \;=\; \left\lfloor \frac{Tf_s - D_{\text{window}} f_s}{s_s} \right\rfloor + 1 \;=\; \Theta\!\left( \frac{Tf_s}{1 - r_{\text{overlap}}} \right)$$

windows per pass.

*Window-level (Alg. 1).* The model produces $B$ window scores and then expands them to a sample mask; both steps are linear in the number of covered samples, giving

$$\mathcal{O}\!\left( \frac{Tf_s}{1 - r_{\text{overlap}}} \right).$$

Accurate onset/duration labeling typically uses a high overlap (e.g., $r_{\text{overlap}} \approx 0.8$), which enlarges the constant by the factor $\frac{1}{1 - r_{\text{overlap}}}$; practical post-processing remains linear and does not change the order.

*Time-step (Alg. 2).* The network outputs one score per sample; thresholding plus light post-processing costs

$$\mathcal{O}(Tf_s) \;+\; \mathcal{O}(D_{\text{window}} f_s) \;=\; \mathcal{O}(Tf_s).$$

Setting $r_{\text{overlap}} = 0$ eliminates redundant evaluations entirely.

# G ARCHITECTURE DETAILS

Table 13: Model design for seizure detection

| Input Size | Operator | kernel / pool | stride | padding |
|---|---|---|---|---|
| $19 \times 15360$ | Conv1d ($19 \to 32$) + ELU | 11 | 1 | 5 |
| $32 \times 15360$ | MaxPool1d | 2 | 2 | 0 |
| $32 \times 7680$ | Conv1d ($32 \to 64$) + ELU | 9 | 1 | 4 |
| $64 \times 7680$ | MaxPool1d | 2 | 2 | 0 |
| $64 \times 3840$ | Conv1d ($64 \to 128$) + ELU | 7 | 1 | 3 |
| $128 \times 3840$ | MaxPool1d | 2 | 2 | 0 |
| $128 \times 1920$ | Conv1d ($128 \to 256$) + ELU | 7 | 1 | 3 |
| $256 \times 1920$ | MaxPool1d | 2 | 2 | 0 |
| $256 \times 960$ | Conv1d ($256 \to 512$) + ELU | 5 | 1 | 2 |
| $512 \times 960$ | MaxPool1d | 2 | 2 | 0 |
| $512 \times 480$ | ResCNNStack | – | – | – |
| $512 \times 480$ | PositionalEncoding + Transformer | – | – | – |
| $512 \times 480$ | Upsample ($\times 2$) | – | – | – |
| $512 \times 960$ | Conv1d ($512 \to 512$) + ELU | 3 | 1 | 1 |
| $512 \times 960$ | Upsample ($\times 2$) | – | – | – |
| $512 \times 1920$ | Conv1d ($512 \to 256$) + ELU | 5 | 1 | 2 |
| $256 \times 1920$ | Upsample ($\times 2$) | – | – | – |
| $256 \times 3840$ | Conv1d ($256 \to 128$) + ELU | 5 | 1 | 2 |
| $128 \times 3840$ | Upsample ($\times 2$) | – | – | – |
| $128 \times 7680$ | Conv1d ($128 \to 64$) + ELU | 7 | 1 | 3 |
| $64 \times 7680$ | Upsample ($\times 2$) | – | – | – |
| $64 \times 15360$ | Conv1d ($64 \to 32$) + ELU | 7 | 1 | 3 |
| $32 \times 15360$ | Conv1d ($32 \to 1$) | 11 | 1 | 5 |
| $1 \times 15360$ | Squeeze (remove channel) | – | – | – |

Table 14: Model design for sleep stage classification

| Input Size | Operator | kernel / pool | stride | padding |
|---|---|---|---|---|
| $2 \times 7680$ | Conv1d ($2 \to 16$) + ELU | 11 | 1 | 5 |
| $16 \times 7680$ | MaxPool1d | 2 | 2 | 0 |
| $16 \times 3840$ | Conv1d ($16 \to 32$) + ELU | 9 | 1 | 4 |
| $32 \times 3840$ | MaxPool1d | 2 | 2 | 0 |
| $32 \times 1920$ | Conv1d ($32 \to 64$) + ELU | 7 | 1 | 3 |
| $64 \times 1920$ | MaxPool1d | 2 | 2 | 0 |
| $64 \times 960$ | Conv1d ($64 \to 128$) + ELU | 7 | 1 | 3 |
| $128 \times 960$ | MaxPool1d | 2 | 2 | 0 |
| $128 \times 480$ | ResCNNStack | – | – | – |
| $128 \times 480$ | PositionalEncoding + TransformerEncoder | – | – | – |
| $128 \times 480$ | Upsample ($\times 2$) | – | – | – |
| $128 \times 960$ | Conv1d ($128 \to 128$) + ELU | 3 | 1 | 1 |
| $128 \times 960$ | Upsample ($\times 2$) | – | – | – |
| $128 \times 1920$ | Conv1d ($128 \to 64$) + ELU | 5 | 1 | 2 |
| $64 \times 1920$ | Upsample ($\times 2$) | – | – | – |
| $64 \times 3840$ | Conv1d ($64 \to 32$) + ELU | 5 | 1 | 2 |
| $32 \times 3840$ | Upsample ($\times 2$) | – | – | – |
| $32 \times 7680$ | Conv1d ($32 \to 16$) + ELU | 7 | 1 | 3 |
| $16 \times 7680$ | AttentionPooling | – | – | – |
| 16 | Linear ($16 \to 5$) | – | – | – |
| 5 | Softmax | – | – | – |

Table 15: Model design for pathological detection

| Input Size | Operator | kernel / pool | stride | padding |
|---|---|---|---|---|
| $23 \times 2000$ | Conv1d ($23{\rightarrow}64$) + ELU | 11 | 1 | 5 |
| $64 \times 2000$ | MaxPool1d | 2 | 2 | 0 |
| $64 \times 1000$ | Conv1d ($64{\rightarrow}128$) + ELU | 9 | 1 | 4 |
| $128 \times 1000$ | MaxPool1d | 2 | 2 | 0 |
| $128 \times 500$ | ResCNNStack | – | – | – |
| $128 \times 500$ | PositionalEncoding + Transformer | – | – | – |
| $128 \times 500$ | Upsample ($\times 2$) | – | – | – |
| $128 \times 1000$ | Conv1d ($128{\rightarrow}128$) + ELU | 3 | 1 | 1 |
| $128 \times 1000$ | Upsample ($\times 2$) | – | – | – |
| $128 \times 2000$ | Conv1d ($128{\rightarrow}64$) + ELU | 5 | 1 | 2 |
| $64 \times 2000$ | AttentionPooling | – | – | – |
| 64 | Linear ($64{\rightarrow}1$) | – | – | – |
| 1 | Sigmoid | – | – | – |

## H  DATASET DESCRIPTION AND DATA PROCESSING

### H.1  SEIZURE DETECTION

- **Siena Scalp EEG Database** Detti (2020): This database contains EEG recordings from 14 patients collected at the Neurology and Neurophysiology Unit of the University of Siena. The cohort includes 9 male participants aged 25 to 71 and 5 female participants aged 20 to 58. Recordings were conducted using Video-EEG at a sampling rate of 512 Hz, with electrode placement following the international 10-20 system. In most cases, 1 or 2 EKG channels were also recorded. An experienced clinician diagnosed epilepsy and classified seizure types based on the standards of the International League Against Epilepsy, following a detailed evaluation of each patient's clinical and electrophysiological data.

- **TUH EEG Seizure Corpus v2.0.3** Shah et al. (2018): This dataset is a curated subset of the TUH EEG Corpus, originally collected from archived clinical EEG records at Temple University Hospital between 2002 and 2017. It includes recordings that were selected based on clinical documentation and the results of seizure detection algorithms to ensure a higher likelihood of seizure presence. Version 2.0.0 features 7,377 EDF files from 675 patients, totaling 1,476 hours of EEG data. The recordings are generally short, averaging around 10 minutes each. The dataset features variability in both sampling rates and the number of EEG channels, though all recordings have a minimum sampling rate of 250 Hz and include at least 17 EEG channels following the 10-20 electrode placement system. Seizure annotations are provided in CSV format, detailing the start and end times, affected channels, and seizure types.

- **SeizeIT1** Vandecasteele et al. (2020): This dataset was collected during the ICON project (2017–2018) in collaboration with KU Leuven and other institutions. It focuses on developing a seizure monitoring system using behind-the-ear (bhE) EEG electrodes, aiming to balance seizure detection accuracy with patient wearability in home environments. Data were recorded during presurgical evaluations in a hospital setting, where patients were continuously monitored via video EEG (vEEG) over several days. A total of 82 patients participated, with 54 having bhE EEG recordings. Among them, 42 patients experienced seizures, yielding between 1 to 22 seizures per patient (median: 3). Available data per patient include full 10-20 scalp EEG, bhE EEG, and single-lead ECG (typically lead II).

- **Dianalund** Dan et al. (2024): The dataset was gathered at the Epilepsy Monitoring Unit (EMU) of the Filadelfia Danish Epilepsy Centre in Dianalund over the period from January 2018 to December 2020, using the NicoletOne™ v44 amplifier. It includes data from 65 patients who experienced at least one seizure during their hospital stay, with each seizure displaying a visually identifiable electrographic pattern on video. In total, 4360 hours of EEG recordings were collected, with patient monitoring durations ranging from 18 to 98 hours. Most participants were adults (median age: 34), and eight were children aged between 5 and 66 years. Across all subjects, 398 seizures were captured and independently annotated by three certified neurophysiologists specializing in long-term video-EEG monitoring. When

disagreements arose, a final consensus label was established. All data were anonymized and converted into a BIDS-compliant format using an adapted version of the epilepsy2bids Python tool tailored for this dataset. EEG recordings were standardized to the 19-channel 10-20 system, re-referenced to a common average, and resampled at 256 Hz.

**Pre-processing.** We arrange every EEG recording's channels in a consistent sequnce: *[Fp1-F3, F3-C3, C3-P3, P3-O1, Fp1-F7, F7-T3, T3-T5, T5-O1, Fz-Cz, Cz-Pz, Fp2-F4, F4-C4, C4-P4, P4-O2, Fp2-F8, F8-T4, T4-T6, T6-O2]*. The signals are then resampled to a common 256 Hz using the Fourier method Virtanen et al. (2020). An Gaussian normalization to each channel is then implemented by calculating

$$x_i^* = (x_i^* - \bar{x})/s_x,$$

$$\bar{x} = \frac{1}{K}\sum_{i=1}^{K} x_i,$$

$$s_x = \frac{1}{K-1}\sum_{i=1}^{K}(x_i - \bar{x})^2.$$

Followed by Zhu & Wang (2023), a bandpass filter was applied to preserve signal components within the 0.5 Hz to 100 Hz frequency range. Following this, two notch filters were used to remove frequencies at 1 Hz and 60 Hz, which commonly correspond to heart rate artifacts and power line interference, respectively. Note that we only use TUSZ v2.0.3, and Siena Scalp EEG is used for dataset formulation. The other two datasets are merely used for cross-dataset evaluation.

## H.2    Sleep Stage Classification

- **Sleep-EDFx** Kemp et al. (2000): This dataset comprises 197 (78 healthy subjects) whole-night polysomnographic (PSG) recordings collected from healthy subjects and individuals with mild sleep difficulties. The recordings include EEG (from Fpz-Cz and Pz-Oz electrode placements), horizontal EOG, submental chin EMG, and event markers. Some records also contain respiration and body temperature measurements. Each PSG recording is accompanied by a hypnogram annotated by trained technicians according to the 1968 Rechtschaffen and Kales manual, detailing sleep stages W, R, 1, 2, 3, 4, movement time (M), and unscored segments.

**Pre-processing.** The preprocessing approach followed the method proposed by EEGPT Wang et al. (2024). Initially, the EEG signals were converted to millivolts (mV). A 30 Hz low-pass filter was then applied to remove high-frequency noise. The recordings were segmented into non-overlapping 30-second windows, and each window underwent z-score normalization independently for each channel.

## H.3    Pathological(abnormal) Detection

- **TUH Abnormal EEG Corpus v3.0.1** Shah et al. (2018): TUAB is a collection of EEG recordings from Temple University Hospital, labeled as either normal or abnormal. It includes 2,993 EEG files recorded between 2002 and 2017. The data is split into training and evaluation sets, with no overlap in patients between them. The training set has 2,717 files from 2,130 people, and the evaluation set has 276 files from 253 people. The formulated dataset contains a total of 409455 10-second samples.

**Pre-processing.** We first removed non-EEG channels, such as EKG, EMG, and respiration were first removed. Next, only recordings with 21 standard 10-20 EEG channels were retained and reordered to a consistent reference montage. Recordings not matching the expected channel order were excluded.

Signals were then bandpass filtered between 0.1 Hz and 75 Hz to remove slow drifts and high-frequency noise. A notch filter at 50 Hz was applied to suppress power line interference. Data were downsampled to 200 Hz for efficiency. Each recording was then segmented into non-overlapping 10-second windows (2,000 samples per segment), and each segment was saved with a label indicating whether it was from an abnormal (1) or normal (0) EEG.

