# OpenReview forum: "Large EEG-U-Transformer for Time-Step-Level Detection Without Pre-Training"
_ICLR.cc/2026/Conference — ICLR 2026 Conference Withdrawn Submission_

### Official Review · Reviewer_Zb9o · 2025-10-29

**Soundness:** 2
**Presentation:** 2
**Contribution:** 2
**Rating:** 4
**Confidence:** 3

**Summary:**

This paper proposes EEG-U-Transformer, a U-shaped architecture that combines convolutional encoders, ResCNN stacks, and transformer modules for time-step-level EEG analysis. The model aims to eliminate sliding window-based post-processing and pre-training requirements, achieving competitive results on seizure detection, sleep staging, and pathological detection tasks.

**Strengths:**

The sequence-to-sequence design effectively bypasses redundant window-level inference, simplifying pipelines for event-centric tasks like seizure detection.

The model scales competitively with large pre-trained foundation models while relying solely on downstream data, reducing computational costs.

**Weaknesses:**

The emphasis on "no pre-training" as a virtue seems misleading; large-scale pre-training learns generalizable representations that often enhance performance and robustness when fine-tuned to specific tasks. By forgoing this, the model may underutilize broad EEG patterns, limiting its adaptability to diverse domains or low-data scenarios. Please note that the data output by different EEG devices can vary significantly, and strong general knowledge is a necessary tool to bridge this gap in clinical practice.

Cross-dataset results (Table 6) reveal significant performance drops (e.g., ~34% F1-score decrease on SeizeIT1), indicating sensitivity to domain shifts in electrodes, demographics, or devices.

The post-processing pipeline (thresholding, morphological operations) remains heuristic and dataset-dependent, undermining the end-to-end promise.

Hyperparameter sensitivity (e.g., kernel sizes, transformer layers) is underexplored, raising reproducibility concerns without meticulous tuning.

Computational analysis focuses on inference time but omits training costs and memory footprint, especially for long sequences (e.g., T=15360).

Global-local interaction and fusion have been extensively explored in numerous previous works, encompassing both the general visual recognition and EEG analysis domains, e.g, [1-4]. Consequently, this cannot be considered a significant contribution of this paper. Furthermore, I did not observe any relevant discussions about these works.

Refs:
[1] On the integration of self-attention and convolution. Arxiv '23.
[2] TransXNet: Learning Both Global and Local Dynamics with a Dual Dynamic Token Mixer for Visual Recognition. Arxiv '23.
[3] Multiscale global prompt transformer for EEG-based driver fatigue recognition. TASE'24.
[4] Learning robust global-local representation from EEG for neural epilepsy detection. TAI '24.

**Questions:**

My questions have been listed in the weaknesses part.

---

> ### Author Response · Authors · 2025-11-21
> **[Part 1/2] Author Response**
>
> ## W1 \- Pretrain
>
> Yes, we agree that the goal of large-scale pre-training is to learn generalizable representations and to enhance performance and robustness on downstream tasks. Indeed, such a process shows tremendous success in fields like Natural Language Processing and Computer Vision. However, EEG analysis is significantly different than these areas. For example, unlike natural languages that have sufficient data to pre-train, EEG recordings are much rarer and their information is much sparser than text, as one recordings contain lots of background noise. **Therefore, we argue that our work is meaningful as our experiments revealed that a well-designed, simple architecture can outperform existing EEG foundation models in terms of accuracy and robustness without reliance on complex pre-training or massive data resources in the EEG analysis area**. Specifically,
>
> * In the Sleep classification and pathological detection task, we conducted a strictly subject-level blind evaluation where the testing set is completely departs from the training set, not only with different waveforms, but also derived from different subjects of different ages, heights, etc. Under such a blind test setting, our model achieves the highest accuracy, indicating the strongest performance and robustness.
> * We further report more baselines’ cross-device performance, including several foundation models in the Dianalund dataset using the latest version of SzCore \[1\] \[2\] benchmark in `Table 1`, where our model also showed the strongest performance. The full cross-device table can be found in the revised paper’s `Appendix C.1`.
>
> **This justified our model design and leads to an under-explored question: whether the pre-training process can exert its power in EEG analysis, especially considering the current constrained data collections**. Understanding this is important because it challenges a common assumption that “pre-training is always beneficial” and instead invites the community to re-examine when and how large-scale representation learning should be deployed in biosignal domains. If pre-training cannot exert its full power in the EEG area, then overly complex pre-training pipelines may provide diminishing returns relative to carefully engineered architectures directly optimized for accurate and robust inference.
>
> | Model | Backbone | F1-score | Sensitivity | Precision |
> | :---: | :---: | ----- | ----- | ----- |
> | EEG-U-Transformer | UNet & ResNet & Transformer | 0.4238 | 0.3693 | 0.4448 |
> | HySEIZa | Hyena hierarchy | 0.1419 | 0.7150 | 0.0975 |
> | SD2025 | LaBraM | 0.0477 | 0.6975 | 0.0271 |
> | NE Illusion | JEPA | 0.0269 | 0.6888 | 0.0157 |
> | STORM | BENDR | 0.0203 | 0.9887 | 0.0112 |
>
> **Table 1: Performance comparison between our model and pre-training-based models in cross-device seizure detection performance, where our model shows the highest performance, indicating its robustness and generalization ability.**
>
> ---
>
> \[1\] Dan, Jonathan, et al. "SzCORE: A Seizure Community Open-source Research Evaluation framework for the validation of EEG-based automated seizure detection algorithms." arXiv preprint arXiv:2402.13005 (2024).
>
> \[2\] Dan, Jonathan, et al. "SzCORE as a benchmark: report from the seizure detection challenge at the 2025 AI in Epilepsy and Neurological Disorders Conference." arXiv preprint arXiv:2505.18191 (2025).

---

> ### Author Response · Authors · 2025-11-21
> **[Part 2/2] Author Response**
>
> ## W2 \- Cross-Device Evaluation
>
> Yes, the performance of models trained over TUSZ drops in SeizeIT1 and Dianalund datasets. However, please note that, as shown in `Table 1,` the dropping performance happened for every model, including large foundation models that were supposed to achieve better generalization and robustness.
>
> We believe that this performance drop is reasonable as these datasets are significantly different than the training set(TUSZ). **For example, the Dianalund dataset was recorded with the NicoletOne v44 amplifier, which was a portable device that allowed patients to move freely within the building**. Distinctions like this lead to noise and unique attributes that are not present in the training corpus.
>
> ## W3\&W4 \- Post-processing & Hyperparameter
>
> The post-processing is designed based on the common seizure activities’ behavior, so it should be able to adapt to classic EEG recordings. In the cross-device experiment, we used the same post-processing procedure, and our method still keeps the strongest performance.
>
> We have also provided the ablation study for the post-processing in the paper’s `Appendix D`. One thing in the post-processing that will be dataset-dependent is the threshold. We have also provided analysis for the threshold in `Appendix D` and have detailly explained how to adjust it based on different scenarios.
>
> Similarly, we have provided hyper-parameter analysis, including hyperparameters for the ime-step level dataset formulation and model-related hyperparameters in `Appendix D` for tuning guidance.
>
> ## W5 \- Computational Analysis
>
> As the motivation for the time-step level classification is to get rid of redundant overlapping inference. Thus, we validate that our method solved such a challenge by providing empirical analysis in the paper’s `Table 3` and theoretical analysis in the paper’s `Appendix F`.
>
> We added the report of the total training time and computational resource usage for each task in the revised version’s `Appendix F`. Specifically,
>
> * TUSZ: Training took 3.1 hours over 60 epochs, with 211 batches, where each batch contains 256 training samples. The training process takes 42130MB of memory.
> * Sleep-EDFx: Training took 4.64 hours over 84 epochs, with 404 batches, where each batch contains 256 training samples. The training process takes 29640MB of memory.
> * TUAB: Training took 2.84 hours over 30 epochs, with 581 batches, where each batch contains 512 training samples. The training process takes 39710MB of memory.
>
> ## W6 \- Related Work
>
> Thank you very much for listing the related work\! As we have clarified, **the proposed model, EEG-U-Transformer, is merely a side contribution of this paper**. More novelty clarification is discussed in our general response of `Novelty Clarification` and `Model Design Justification`. We have added these papers to the paper’s `Related Work` section.

---

> > ### Comment · Reviewer_Zb9o · 2025-11-26
> >
> > Thank you for your response, which has addressed some of my concerns. Although the authors emphasize that the architecture is not the core contribution, the global-local information aggregation mechanism is a powerful technique that can significantly improve performance in many domains. Therefore, I suggest that the authors provide more evidence to demonstrate the adaptability of their core contribution to different architectures to investigate the impact of architecture on performance. Additionally, while the authors have highlighted that architecture design is a side contribution, it is still heavily emphasized in the original manuscript, potentially leading readers to attribute the performance improvements mainly to this aspect. I also noticed that other reviewers have raised numerous concerns, which has led me to maintain my original rating.

---

> > > ### Author Response · Authors · 2025-12-04
> > > **Thank you, Reviewer Zb9o**
> > >
> > > We thank the reviewer for the constructive feedback.
> > >
> > > Indeed, the global-local information aggregation mechanism is a powerful technique, which is why we design such a model to be integrated into our framework, which, as described above, is also part of our contribution. We have shown that combining the EEG-U-Transformer with the proposed time-step level representation learning framework can outperform baselines across both types of tasks.
> > >
> > > We also agree that benchmarking our framework with more architectures will be helpful to better understand the framework and also justify our model design choices. We have reported the performance of the pure-Transformer and pure-U-Net architectures in `Model Design Justification`.
> > >
> > > We appreciate the feedback regarding the writing emphasis. We will adjust the presentation based on our response to `Novelty Clarification` and `Model Design Justification` to better balance the contributions and ensure that readers clearly understand the primary focus and scope of the work.

---

### Official Review · Reviewer_bx19 · 2025-10-29

**Soundness:** 2
**Presentation:** 1
**Contribution:** 3
**Rating:** 4
**Confidence:** 4

**Summary:**

The paper introduces a new architecture for an EEG-based application. This model uses a U-Net architecture with Transformer layers. This allows to have more generalizable networks. The model is then tested on three different modalities: seizure detection, sleep staging, and abnormality detection.

**Strengths:**

- The authors propose a new architecture that leverages the power of U-Net with the generalization power of the transformer between the encoder and the decoder.
- This model offers versatility in the application for both Windows-based event and segmentation tasks.
- This new model is applied to three different modalities for which the model outperforms other models, except for the TUAB dataset, where the results are worse with their model.
- In addition, the model is faster to run compared to other models.

**Weaknesses:**

If the paper is proposing a new model that works well on several modalities, I think that the paper lacks clarity in some points:
- In Figure 1, it is not so clear that the arrows represent layers, especially when scaling embeddings, and Figure 3 represents layers by blocks.
- In the same Figure, it will be clearer to use a letter instead of a number (example: 15360 -> T and then T/2 ...)
- In Figure 2, the windowing of the signal is exactly the same between the two examples. Since the strength of the method is to be able to cut the windows as we want, it could be interesting to see the possible difference.
- The organization of the paper is hard to follow sometimes. Part 2.3 gives information on the pre-processing of the dataset, but Part 2 focuses on the architecture of the model. I would suggest moving it to part 3.1 or at least to the end of part 2.
- This claim: "Such experimental settings and
evaluation metrics do not fit with real-world requirements and often limit the model design, as
different model architectures might benefit from different sequence lengths," is not true for all tasks. In sleep staging, for example, all the datasets are annotated every 30 seconds. Adding a reference can give more strength to the claim that is central to the motivation of the paper.

Minor:
- add number of subjects in Table 1
- For the TUAB dataset, the claim that you are in the top tier of the AUROC score and "marginally" lower than BIOT is too strong. Your method is losing 3% compared to Biot, which is the improvement that you have on sleed-EDFx.
- Several typos were seen in the paper.

In my opinion, this paper has good propositions and results, but the lack of clarity makes it hard to follow.

**Questions:**

- You are categorizing the datasets into no-activity, full-activity, and partial-activity. Is it something usual for one of the modalities? Did you do that on every dataset?
- On which dataset was the ablation study done?
- In sleep staging, the models are usually using sequences of windows. For example, in the U-Sleep paper, they are using 35 windows as input. This is giving more context to the models. Is it something doable with your method? For sleep, for example, could we pass a longer time length than 30 seconds to get multiple outputs at the end?
- For the time comparison (Table 3), why are only 3 competitors given? Knowing that you run several models, it would be easy to give every running time.

---

> ### Author Response · Authors · 2025-11-21
> **[Part 1/3] Author Response**
>
> ## W1 \- Figure 1 Arrow
>
> In Figure 1, all arrows correspond to actual layer operations, and we explicitly distinguish them using five different arrow colors, each explained in the legend on the right (Conv1D+ELU, pooling, upsampling, residual connections, and Conv1D+sigmoid). Within the Scaling Embedding module, the internal components are represented using distinct color blocks: the first block corresponds to the ResCNN stack, whose detailed structure is provided in the “Res CNN Stack” panel; the second block corresponds to positional embedding, clearly marked with a textual label; and the final block represents the Transformer encoder, shown as a dedicated “Transformer-Encoder Stack.”
>
> ## W2 \- Figure 1 Sequence Notation
>
> Thank you very much for the advice\! We have modified our graph to use letters to demonstrate the sequence length and channel number. You can find the change in the revised paper.
>
> ## W3 \- Figure 2 Segmentation Example
>
> Thank you for the helpful suggestion\! We have modified the figure to add two different model examples to show that models can pick their own sequence length to maximize their performance.
>
> ## W4 \- Organization
>
> We have moved the sequence-to-sequence modeling section to the end of Part 2(Methodology) and ensured that the model architecture parts(EEG-U-Transformer and Attention Pooling) are together. We choose not to move to Part 3(experiments) as such a modeling framework is also part of our methodology contribution.
>
> ## W5 \- Claim Reference
>
> Thanks for mentioning\! Here, we are specifying event-centric(time-step level classification) tasks, which, in this paper, is the seizure detection task. We added this specification in the paper and have added some related references next to this claim.

---

> ### Author Response · Authors · 2025-11-21
> **[Part 2/3] Author Response**
>
> ## Minor Weakness
>
> ### W1 \- Subject Number
>
> We have added the number of subjects to Table 1\.
>
> ### W2 \- TUAB results conclusion
>
> Thank you for pointing this out\! We modified our conclusion to mention that we achieved the best performance in the balanced accuracy metric, but not in the AUROC metric. We also provided a potential explanation for the AUROC result.

---

> ### Author Response · Authors · 2025-11-21
> **[Part 3/3] Author Response**
>
> ## Q1 \- Dataset Formulation
>
> The categorization process is part of our proposed time-step level classification framework to ensure a balanced dataset in the dataset formulation phase. We use such a process in the time-step level classification experiments(seizure detection), but not in window-level tasks(sleep and pathological detection). Instead, in window-level tasks, we strictly followed the most recent window-level works’ experimental setting to conduct comparative experiments under the same framework.
>
> ## Q2 \- Ablation Study
>
> We conduct the ablation study in the TUSZ dataset for the time-step level classification task.
>
> ## Q3 \- Sleep Staging & Longer Sequence
>
> Thanks for mentioning\! In fact, that is what we did in our sleep stage classification, where bags of 30s windows are fed into the model. **We use 30s to be the window size because we followed EEGPT’s setting to conduct comparative experiments**.
>
> We have tried different sequence lengths in the time-step level classification task, as shown in the table reported in the general response of `Model Design Justification,` where it shows that **our model performance improves as the sequence length increases from 10s to 60s**.
>
> ## Q4 \- Runtime
>
> `Table 3` is the runtime efficiency comparison between the time-step level model and the window-level model, which are the baselines listed in the table.

---

> > ### Comment · Reviewer_bx19 · 2025-11-27
> >
> > I thank the reviewer for the answers. I still have concerns about the paper, primarily regarding its presentation. The paper would benefit from a clearer presentation. For that, I'll maintain my score.

---

> > > ### Author Response · Authors · 2025-12-04
> > > **Thank you, Reviewer bx19!**
> > >
> > > We thank the reviewer for the feedback regarding the clarity of the presentation. In the revised version, we will reorganize and refine the narrative to improve readability and ensure that the key contributions are communicated more clearly.

---

### Official Review · Reviewer_qfhj · 2025-10-30

**Soundness:** 2
**Presentation:** 2
**Contribution:** 2
**Rating:** 2
**Confidence:** 4

**Summary:**

This paper proposes the Large EEG-U-Transformer (EEG-U-T), a sequence-level model designed to perform end-to-end EEG time-step detection without pre-training. The authors argue that existing EEG foundation models (e.g., BIOT, LaBraM, EEGPT) rely on fixed window segmentation and heavy pre-training, leading to redundant processing and computational overhead. To address this, they introduce a U-shaped CNN–Transformer encoder–decoder that directly models long EEG sequences in a sequence-to-sequence manner, aiming to capture both local and global temporal dependencies. The model is evaluated on three datasets (TUSZ, TUAB, Sleep-EDF) across seizure detection, pathological EEG classification, and sleep staging tasks.

**Strengths:**

The paper tackles an important and timely problem: improving EEG temporal modeling beyond single-epoch representations. The paper is easy to follow with a clearly described architecture.

**Weaknesses:**

IMO the paper has a mispositioned motivation and runs the risk of overclaiming. It mischaracterizes the limitations of existing foundation models such as BIOT, LaBraM, and EEGPT. Those models are not designed for sequence-to-sequence detection but focus on learning single-epoch representations and channel adaptation, which are foundational rather than temporal tasks. The proposed model instead targets continuous sequence labeling—a fundamentally different objective. Moreover, the claim that existing foundation models suffer from “window segmentation limitations” is inaccurate and overstates their weaknesses. Windowing is a task-dependent design choice, not a methodological flaw.

Moreover, I am afraid the novelty is limited in that the proposed method is essentially a sequence wrapper over existing epoch-based models. The proposed EEG-U-Transformer can be viewed as stacking a CNN–Transformer encoder-decoder to model long-range temporal dependencies. Conceptually, this is a sequence-level wrapper built upon the same features that existing single-epoch models already extract effectively. It would be straightforward to use features from published foundation models (BIOT, LaBraM, EEGPT) and train a small seq-to-seq model for fine-tuning. Hence, the contribution lies more in architectural repackaging than in a novel modeling principle or learning paradigm.


The authors claim that existing foundation models require “diverse datasets and tremendous computation resources.” However, the proposed EEG-U-T introduces much larger model size (Table 5) and a heavier training process, contradicting the stated motivation of efficiency. While parameter counts increase significantly, the AUROC actually drops below BIOT on TUAB. Table 3 compares the proposed model’s runtime against task-specific baselines. However, the proposed method is presented as a foundation-style model, which in realistic usage would employ only the final-layer embeddings for inference. Thus, comparing full training runtime across architectures with different usage paradigms is not meaningful. A fair comparison would include inference-only latency and pre-training vs. fine-tuning cost under identical settings.

Regarding experiments, the setup lacks uniformity and rigor:
- Different baselines are used across datasets (TUSZ vs. TUAB vs. Sleep-EDF), making comparisons non-standardized.
- EDF is a very small dataset—training a large model on such a dataset raises questions about overfitting and necessity.
- TUAB results differ from reported metrics in prior papers, yet no explanation or setting alignment is provided.
- No parameter sensitivity, ablation, or standard deviation analysis is presented.
Overall, the empirical evidence is not strong enough to support the claimed generalization or efficiency improvements.

I should also point out that the contribution is incremental: recent research on EEG foundation models have already explored representation learning, cross-channel dynamics, and scaling. This paper does not introduce a fundamentally new modeling principle—it mainly combines known CNN–Transformer and U-Net design patterns for sequence labeling. The contribution feels incremental, and the conceptual advancement over existing frameworks remains limited.

**Questions:**

please refer to the weaknesses.

---

> ### Author Response · Authors · 2025-11-21
> **[Part 1/4] Author Response**
>
> ## W5 \- Incremental Contribution
>
> Please see our general response for `Novelty Clarification` and `Model Design Justification.`
>
> ## W1 \- Foundation Model’s Limitation
>
> ### Different Objective
>
> We completely agree that “**foundation models are not designed for sequence-to-sequence detection, but for single-epoch representations**”. In fact, in this work, we are not arguing about their capability on seq2seq detection or comparing with them on such tasks. Instead, as mentioned in the general response of `Novelty Clarification`, **we are arguing that our unified framework not only achieves efficient and impressive time-step level performance, but also maintains the strong ability in window-level classification tasks that are comparable to existing window-level foundation models**. To validate that, we reproduced the published foundation models’ experiments and compared their classification performance with ours, and showed that our work achieves the best accuracy performance.
>
> ### Window Segmentation Limitations
>
> We are not sure what you refer to, as we did not mention “window segmentation limitations” in the papers. We listed some points that might be helpful.
>
> * We mentioned that existing foundation models are pre-trained over a short window with a small sequence length. This is not good, as a lot of target events, such as seizure activities, often span several minutes; thus, pre-training over small windows cannot help. More detail can be found in the response of `W2 - Why not use Features from Existing Foundation Models`.
> * We mentioned that window-level classification requires redundant overlapping window inference to map discrete labels back to annotation masking that describes the target event’s onset and duration time. In contrast, our time-step level classification model does not need such redundant overlapping inference as it directly outputs annotation predictions.

---

> ### Author Response · Authors · 2025-11-21
> **[Part 2/4] Author Response**
>
> ## W2 \- Why not use Features from Existing Foundation Models
>
> Thank you very much for the thoughtful advice\! We do not directly use features from published foundation models because of the following considerations:
>
> * This research work starts with an application to the seizure detection task. However, **Most existing foundation models are pre-trained with the use of a small window size**. For example, EEGPT is pre-trained over windows with a sequence length of 1024 time steps, sampling from 256Hz. **In contrast, lots of target events in EEG analysis, including but not limited to seizure activities, often span several minutes with more than 10000 time-steps**. Therefore, we argue that it is not the best choice to directly use the feature from existing foundation models. As a result, we chose to propose a U-shaped model to take a long sequence input(*1-minute* window with 15360 sequence length) and efficiently downsample it through convolution layers. **More model design explanations are provided in the general response on `Model Design Justification`.**
> * We empirically validated such an intuition by integrating EEGPT into our framework as a baseline in the seizure detection task and reported in `Table 1`. As shown in the table, due to the short pre-training context window, **EEGPT underperforms our proposed EEG-U-Transformer with a low F1-score and a low precision. Such a low precision leads to a higher false rate, even with a threshold of 0.9**.
> * Moreover, we argue that such a strong performance is also part of this paper’s novelty, as it revealed insights and an important but under-explored question discussed in the last paragraph of the general response of `Novelty Clarification`.
>
> | Model | Threshold | Event F1 | Sensitivity | Precision |
> | :---: | ----- | ----- | ----- | ----- |
> | EEGPT | 0.9 | 0.5324 | 0.7640 | 0.4085 |
> |  | 0.8 | 0.5014 | 0.8053 | 0.364 |
> |  | 0.7 | 0.4740 | 0.8201 | 0.3333 |
> | EEG-U-Transformer | 0.8 | 0.6713 | 0.7168 | 0.6312 |
>
> **Table 1: Performance comparison between pre-trained EEGPT and EEG-U-Transformer in the seizure detection task, where EEGPT shows a lower F1-score with low precision even under a 0.9 threshold.**

---

> ### Author Response · Authors · 2025-11-21
> **[Part 3/4] Author Response**
>
> ## W3 \- Efficiency/Accuracy Claim
>
> We clarify with the following points to show why it is self-evident that our model achieves a better efficiency/accuracy:
>
> * The most recent foundation model, EEGPT \[1\], introduced a 25M model, which is three times bigger than ours.
> * **Foundation models require extensive pre-training before finetuning on downstream tasks.** Such a process requires a large amount of data and much more training time. In contrast, our model does not need the pre-training process to achieve strong performance.
> * We acknowledge that the AUROC results in TUAB are not the best ones, and have provided our assumption regarding the failure of this metric in the paper’s `Section 3.3`. **On the other hand, please note that we have achieved the highest *Balanced Accuracy* in TUAB and demonstrate the state-of-the-art performance across other tasks(seizure detection and sleep stage classification) in every metric**. It is not fair to solely look at one metric’s failure but ignore all other metrics’ success.
> * `Table 3` actually is the evaluation of “realistic usage runtime” that the reviewer asked for instead of “meaningless training runtime”. In fact, the motivation of time-step level classification is to get rid of the redundant overlapping post-processing procedure. That is why we compare the realistic runtime efficiency in `Table 3` to validate it.
>
> ---
>
> \[1\] Wang G, Liu W, He Y, et al. Eegpt: Pretrained transformer for universal and reliable representation of eeg signals\[J\]. Advances in Neural Information Processing Systems, 2024, 37: 39249-39280.

---

> ### Author Response · Authors · 2025-11-21
> **[Part 4/4] Author Response**
>
> ## W4
>
> ### Different Baselines
>
> **It is inevitable to use different baselines across different tasks because different models are proposed against different tasks**. For example, EventNet \[1\] is a model that was designed for the seizure detection task only, so it is inappropriate to compare it with this model in the sleep classification task. In that case, we used the most recent and previously state-of-the-art models to be our baselines for each task and **ensured that every task was compared in a standardized manner**.
>
> ### EDF Dataset
>
> **The EDF dataset is used by EEGPT in the sleep classification task. We followed their experimental setting** to conduct the experiment. It is also noteworthy that, following EEGPT, we split the training/test set at the subject level, which makes the testing set completely blind to the training set.
>
> ### TUAB Result
>
> We believe that our results align with EEGPT’s report. We also open-sourced our code anonymously, and every experiment is reproducible.
>
> ### Ablation Analysis
>
> We have provided an ablation study of the model in `Section 3.2`, and provided sensitivity across various hyperparameters in `Appendix D`. The experiments are repeated five times with different random seeds, and we have reported the standard deviation in the experiment section.
>
> ---
>
> [1] Seeuws, Nick, Maarten De Vos, and Alexander Bertrand. "Avoiding post-processing with event-based detection in biomedical signals." IEEE Transactions on Biomedical Engineering 71.8 (2024): 2442-2453.

---

### Official Review · Reviewer_btKw · 2025-11-01

**Soundness:** 3
**Presentation:** 3
**Contribution:** 2
**Rating:** 4
**Confidence:** 3

**Summary:**

This paper introduces the "Large EEG-U-Transformer," a hybrid U-Net and Transformer architecture. It is designed for efficient, time-step level  event detection. The paper's central and most important claim is that this model, trained only on downstream datasets, can outperform large, pre-trained foundation models (LFMs) like EEGPT and BIOT.

**Strengths:**

1/ The core result—that a 6.1M parameter model trained from scratch can beat a 25M pre-trained LFM (EEGPT) on Sleep-EDFx.
Strong Empirical Performance: The model achieves convincing SOTA results on the TUSZ seizure task (beating DeepSOZ-HEM)。and the Sleep-EDFx task. And the algorithm is of high efficiency compared with prior arts.

**Weaknesses:**

The claim in Appendix A that combining U-Nets and Transformers is novel for biomedical signals is false. This concept is foundational in medical imaging (e.g., UNETR, Swin-Unet)  and exists in time-series (Yformer). The authors also failed to cite highly relevant prior work using attention-gated U-Nets for the exact same task (Chatzichristos et al. 2020).

The paper's scaling study (Tables 9 & 10) is a weakness, not a strength. Performance peaks at 59.9M parameters and then drops significantly at 80.9M . This contradicts the scaling laws that underpin LFMs and is poorly explained.

**Questions:**

See the weakness part.

---

> ### Author Response · Authors · 2025-11-21
> **Author Response**
>
> ## W1 \- Novelty
>
> Please see our general response to `Novelty Clarification` and `Model Design Justification`, where we detailed explained our main novelty and why we use such an architecture in our proposed time-step level classification framework.
>
> For the related work, we believe that we have cited the paper that the reviewer mentioned(Chatzichristos et al. 2020\) right in `Appendix A`.
>
> ## W2 \- Model Size Ablation Study
>
> We have explicitly mentioned the accuracy drop and provided a potential explanation in the scaling study(`Appendix D`).
>
> Specifically, unlike areas like Natural Language Processing, where there exists sufficient data for model training. EEG recording corpus are much rarer than text, which limits the dataset size. Therefore, at a certain point, the model will be too large to learn from a restricted training sample.

---

### Official Review · Reviewer_UH5h · 2025-11-01

**Soundness:** 3
**Presentation:** 2
**Contribution:** 2
**Rating:** 4
**Confidence:** 4

**Summary:**

The paper couples a U-Net style temporal convolutional backbone with a Transformer encoder to build a sequence-to-sequence model for per-time-step EEG labeling, and then uses attention pooling to convert those features into window-level predictions for tasks like seizure detection and sleep/abnormal screening. On TUSZ it reports solid event-level F1 and fast inference, but the method and evaluation largely combine standard components, so originality is limited.

The major contributions include:
1. A unified EEG framework that performs time-step segmentation and window-level classification in one model, using U-Net temporal features, Transformer encoding, and attention pooling to bridge granular and aggregate predictions.
2 An efficient inference and post-processing pipeline that achieves competitive event-level performance on TUSZ while keeping runtime low, demonstrating practical viability for long recordings.

**Strengths:**

Originality: One backbone handles both per-time-step labeling and window classification via attention pooling.
Quality: Clinically aligned time-step and event-level metrics plus simple post-processing reduce false alarms.
Clarity: Figures and equations clearly explain the architecture and attention pooling.
Significance: Competitive TUSZ event-level F1 with fast inference shows practical value on long recordings.

**Weaknesses:**

1. Limitted novelty: U-Net–style temporal conv plus a Transformer encoder is already common in time-series/biomed (see recent EEG/ECG segmentation hybrids); the paper does not show why this variant is fundamentally better than a strong pure-Transformer or pure-UNet baseline under identical settings.

2 I am worry that authors made several Unfair or under-specified comparisons. Several baselines use different window lengths, channels, or preprocessing, so current tables cannot isolate gains from the proposed model rather than from setup differences; a “same data, same window, same channels, same hardware” table is missing.

3 Mathematical/formulation glitches. The time-step loss mixes indices and does not clearly sum over dataset/time; positional encoding uses a nonstandard denominator likely to be a typo, which hurts clarity and reproducibility.

4 Event-level evaluation is too forgiving. Results rely on tolerant matching and post-processing (morphological ops, min-duration) without reporting FP/h, onset/offset error, or sensitivity to threshold, so it is unclear whether the method is robust in stricter clinical regimes.

Additionally, figures/tables not presentation-ready. Table 1 mixes dataset stats with model configs, and several figures lack legend/abbreviation expansion, making it hard to verify the pipeline or reproduce it.

**Questions:**

1 You use 10000^(2i/Td) rather than the standard 10000^(2i/dmodel). Is this intentional? Please justify the choice and provide an ablation/sensitivity study versus the standard form.
2 Can you re-train and re-evaluate all baselines and your model under identical settings (same channels, window length, preprocessing, hardware, and batch size), reporting mean ± 95% CI over multiple seeds?
3 Current results rely on tolerant matching and post-processing. Please report FP/h, onset/offset error distributions, and FROC, include tolerance/threshold sweeps, and ablate the morphological filtering and minimum-duration pruning to quantify their contribution.
4 could you provide Leakage control with overlapping windows? It would be fair if authors could precisely document train/val/test split policy (file/patient level) and show that overlapping windows do not cross splits. Provide a sensitivity analysis to different overlap ratios to rule out temporal leakage.

**Details Of Ethics Concerns:**

Privacy and safety: Provide IRB/DUA IDs for each dataset, confirm complete PHI removal and redistribution rights, and add a clear “not for clinical use” disclaimer in the release.

Responsible research practice: List dataset versions and approvals, state whether new labels were created with annotator qualifications and compensation, and prove patient-level isolation across train/val/test splits.

Fairness: Report subgroup performance by age, sex, site/device, and comorbidities with confidence intervals to detect and quantify disparate error rates.

---

> ### Author Response · Authors · 2025-11-21
> **Author Response**
>
> ## W1 \- Limited Novelty
>
> Please see our general response to `Novelty Clarification` and `Model Design Justification.`
>
> ## W2\&Q2 \- Comparative Experiments
>
> In the seizure detection task,
>
> * We used the optimal hyper-parameter set proposed by baseline papers. Changing them will only make performance worse, as described by the ablation study in their papers.
> * Letting baseline pick its best hyper-parameter set is a common setting in the seizure detection task, as baselines’ frameworks are significantly different from each other. Such a setting is seen in related work \[1\] \[2\] \[3\] \[5\]. In fact, our experiments are directly employed from the SzCore benchmark \[1\] \[5\].
>
> In sleep staging and pathological detection experiments,
>
> * We followed the most recent work(EEGPT \[4\]) to use the same hyperparameters, including but not limited to pre-/post-processing procedure, channel number, and sequence length.
> * We show that our model outperforms every baseline in these comparative experiments.
>
> ---
> \[1\] Dan, Jonathan, et al. "SzCORE: A Seizure Community Open-source Research Evaluation framework for the validation of EEG-based automated seizure detection algorithms." arXiv preprint arXiv:2402.13005 (2024).
>
> \[2\] Chatzichristos, Christos, et al. "Epileptic seizure detection in EEG via fusion of multi-view attention-gated U-net deep neural networks." 2020 IEEE Signal Processing in Medicine and Biology Symposium (SPMB). IEEE, 2020\.
>
> \[3\] M. Shama, Deeksha, Jiasen Jing, and Archana Venkataraman. "DeepSOZ: A robust deep model for joint temporal and spatial seizure onset localization from multichannel EEG data." International Conference on Medical Image Computing and Computer-Assisted Intervention. Cham: Springer Nature Switzerland, 2023\.
>
> \[4\] Wang, Guangyu, et al. "Eegpt: Pretrained transformer for universal and reliable representation of eeg signals." Advances in Neural Information Processing Systems 37 (2024): 39249-39280.
>
> \[5\] Dan, Jonathan, et al. "SzCORE as a benchmark: report from the seizure detection challenge at the 2025 AI in Epilepsy and Neurological Disorders Conference." arXiv preprint arXiv:2505.18191 (2025).
>
> ## W3 \- Formula Typo
>
> Thank you for pointing this out. We have solved the typo in the revised version.
>
> ## W4\&Q3 \- Post-processing & Threshold Sensitivity
>
> We have conducted an ablation study of post-processing and have reported our model’s sensitivity to the threshold in the Appendix.
>
> * Specifically, we reported threshold sensitivity in the paper’s `Table 8` and showed a smooth sensitivity/precision trade-off. Higher thresholds favor precision at the expense of sensitivity, whereas lower thresholds improve sensitivity but increase false positives. In practice, the threshold can be tuned to clinical priorities: intensive care monitoring may prioritize sensitivity, while wearable devices benefit from higher precision to reduce alarm fatigue.
> * Additionally, we have reported the post-processing ablation study in the paper’s `Table 11`, which “**quantified the contribution of morphological filtering and minimum-duration pruning**”.
>
> ## Q4 Train/Val/Test Split Policy
>
> * In the seizure detection and pathological detection tasks, the data have been pre-split into training, validation, and evaluation sets by the data providers. In that case, we directly used the downloaded split dataset to train, validate, and test. **The three sets are completely separated at the patient level, which ensures 0% subject/recording overlapping**.
> * In the sleep stage classification and pathological detection task, we strictly followed the most recent work(EEGPT) to formulate a train-val-test set. **The formulated set is also subject-level blind**.

---

> ### Author Response · Authors · 2025-11-21
> **Response to Ethical Concern**
>
> ## Ethical Concern
> The reviewer UH5h  is concerned about the ethical issues with the data sets that we used in our paper. All the data sets have been publicly available as benchmarks to the community for many years. Most recently, these benchmark data sets were used in an international competition \[1\] \[2\].
>
> Asking us to provide IRBs and is unreasonable since we are not the ones who created and provided these benchmark data to the community. Requiring us to  "List dataset versions and approvals, state whether new labels were created with annotator qualifications and compensation, and prove patient-level isolation across train/val/test splits." and demanding us to "Report subgroup performance by age, sex, site/device, and comorbidities with confidence intervals to detect and quantify disparate error rates." is totally unacceptable and hostile. These comments indicate that the reviewer is not familiar with these open-source benchmarks used by the international research community. We will formally complain about this review to the Area Chair and conference organizers.
>
> ---
>
> \[1\] Dan, Jonathan, et al. "SzCORE: A Seizure Community Open-source Research Evaluation framework for the validation of EEG-based automated seizure detection algorithms." arXiv preprint arXiv:2402.13005 (2024).
>
> \[2\] Dan, Jonathan, et al. "SzCORE as a benchmark: report from the seizure detection challenge at the 2025 AI in Epilepsy and Neurological Disorders Conference." arXiv preprint arXiv:2505.18191 (2025).

---

### Author Response · Authors · 2025-11-21
**General Response - Novelty Clarification**

We want to humbly acknowledge that the modules that we used, such as UNet and Transformer, are well-established techniques, and we have listed some related work. **However,  the proposed model, EEG-U-Transformer, is merely a side contribution of this paper. Our main novelty lies in the time-step level representation learning framework and its adaptivity to maintain strong performance in window-level classification tasks.**

**By proposing such a framework, we further propose the EEG-U-Transformer to be integrated into this framework. We show the design philosophy of this model in the general response of `Model Design Justification` to show that the EEG-U-Transformer is suitable for our framework.**

## Unified EEG Analysis Framework
Firstly, in this paper, we pointed out that most existing work focuses on window-level representation learning, which, when it comes to downstream tasks, requires redundant overlapping inference for event-centric tasks. Starting from this motivation, **we go beyond window-level representation and propose a training/inference framework to do sequence-to-sequence modeling**. We show that such a method eliminates the overlapping requirement and achieves:

* a smaller time complexity(Appendix F)
* significant(10-fold) inference efficiency improvement in the simulated experiments,
* where our model can process a 1-hour EEG recording in just 3 seconds.

Moreover, beyond handling time-step level classification for event-centric tasks, we integrate an attention-pooling layer to maintain the model’s ability to do window-level classification for status-centric tasks. **Such a combination makes our method a unified solution for EEG analysis that can handle both event-centric(time-step level) and status-centric(window level) tasks**. And our experimental results showed that our model achieves the state-of-the-art in both types of tasks.

## Model
Last but not least, we propose a simple model architecture that is fit with this unified framework and integrate it into our framework. Although the building blocks are well-established techniques, we justify our design choice in the general response of `Model Design Justification` to show the intuition of employing each module. **As a result of such a design, our method achieves the state-of-the-art performance, which is impactful to the community**.

## Experiments Insights
Beyond justifying that our methodology solved the aforementioned challenges and demonstrating the strongest performance, as discussed in the paper’s `Section 4`, **we argue that our experimental results revealed a meaningful insight that a strong performance can be achieved through a well-designed, simple architecture without reliance on complex pre-training or massive data resources in the EEG analysis area**. Such results lead to an under-explored question: **whether the pre-training process can exert its power in EEG analysis, especially considering the current constrained data collections**.

Understanding this is important because it challenges a common assumption, derived from the success of other fields like NLP that are significantly different from EEG analysis, that “pre-training is always beneficial” and instead invites the community to re-examine when and how large-scale representation learning should be deployed in biosignal domains. If pre-training cannot exert its full power in the EEG area, then overly complex pre-training pipelines may provide diminishing returns relative to carefully engineered architectures directly optimized for accurate and robust inference.

We summarize our clarified novelty at the end of the introduction section of the paper.

---

### Author Response · Authors · 2025-11-21
**General Response - Model Design Justification**

We list the following points to show that **it is a natural idea to design such architecture that is perfectly adaptive to our framework** and to answer the reviewer UH5h’s question: “**Why is this variant fundamentally better than pure-Transformer or pure-UNet?**”:

* We first report the results of our pre-empirical analysis in `Table 1`, where we attempted to deploy a pure Transformer model in our framework. **The pure-Transformer model has achieved only modest performance in every sequence length we tested(10s, 30s, 60s).**
* This is intuitively reasonable as the semantic information of time series is mainly hidden in the temporal variation, for which the self-attention model cannot effectively extract. There is work that has empirically shown that convolution achieves better classification performance than Transformer \[3\].
* Moreover, as shown in `Table 1`, **the amount of memory grows with the sequence length in self-attentive models**. This is because in a self-attention model, for a sequence with length T, each token attends to every other token, which leads to an $O(T^2)$ attention matrix. **This is fatal as lots of target EEG events, such as seizure/epilepsy, span several minutes, which requires a large window size to do effective time-step level representation learning**.
* Because of these two factors, we choose to use convolution layers before the Transformer module to exploit local structures and to down-sample the given long sequence. After the transformer module, as we aim to do time-step level representation learning, we use transpose convolutions to reconstruct to original sequence length. Essentially, this gives a U-shaped neural network.

* We also have intuitively discussed the drawbacks of a pure-UNet architecture in our paper’s `Section 1`: “**U-Net primarily operates within local receptive fields, making it difficult for U-Net to capture global features effectively. Beyond that, building up a U-Net requires stacking deeper layers, often leading to vanishing gradients and overfitting**”.
* We justified our intuitive assumption by reporting its performance in our paper’s ablation study in `Section 3.2`. As shown in `Figure 4`, **the vanilla U-Net has an underwhelming performance with a low AUROC mean**. **Integrating both the ResCNN and Transformer stacks produces not only a higher mean AUROC but also a reduced variance with fewer extreme false cases, indicating that these components complement each other effectively**.

**Reviewer qfhj** also suggests directly using existing foundation models’ features. Here, **we tried integrating a pre-trained EEGPT into our framework to be a new baseline and report performance in `Table 2`, where our model outperformed EEGPT**. More information can be found in the response to Reviewer qfhj.

| Sequence Length | Transformer |  | EEG-U-Transformer |  |
| :---- | ----- | :---- | ----- | :---- |
|  | Memory Usage(batch=4) | F1-score | Memory Usag(batch=4) | F1-score |
| 10s | 1596.1 MB | 0.3503 | 367.0 MB | 0.5839 |
| 30s | 4752.0 MB | 0.4375 | 653.2 MB | 0.6444 |
| 60s | 9482.8 MB | 0.3738 | 1156.2 MB | 0.6713 |
| 120s | 18945.9 MB | Out of Memory | 2077.6 MB | 0.6701 |

**Table 1: Performance comparison between pure-transformer and EEG-U-Transformer in seizure detection task. The memory usage of pure-Transformer significantly increases as the sequence length increases, while the performance drops in longer sequence lengths.**

| Model | Threshold | Event F1 | Sensitivity | Precision |
| :---: | ----- | ----- | ----- | ----- |
| EEGPT | 0.9 | 0.5324 | 0.7640 | 0.4085 |
|  | 0.8 | 0.5014 | 0.8053 | 0.364 |
|  | 0.7 | 0.4740 | 0.8201 | 0.3333 |
| EEG-U-Transformer | 0.8 | 0.6713 | 0.7168 | 0.6312 |

**Table 2: Performance comparison between pre-trained EEGPT and EEG-U-Transformer in the seizure detection task, where EEGPT shows a lower F1-score with low precision even under a 0.9 threshold.**

---

### Note · Authors · 2025-12-04

I have read and agree with the venue's withdrawal policy on behalf of myself and my co-authors.